# Layer-Centric Factors of Variation Disentanglement for Task- and Model-Agnostic Generalization

**Hee-Jun Jung** [1]   **Jongmin Park** [1]   **Minwoo Kang** [1]   **Hoyong Kim** [1]   **Kangil Kim** [1]

## Abstract

Disentanglement learning aims to separate the underlying factors of variation (FoV) to improve generalization. However, most FoV-based latent-vector-centric methods impose objective-driven constraints at a bottleneck, and it is difficult to translate disentanglement into consistent gains on downstream tasks without inductive bias. Motivated by architectural approaches complementary to vector-centric objectives for downstream tasks, we propose the *Orthogonal Subspaces Projection* (OSP) layer, a plug-and-play module that integrates into intermediate layers and promotes FoV separation by projecting latent features into mutually orthogonal subspaces. Across diverse domains and tasks, models equipped with the OSP layer improve disentanglement quality and generalization in downstream tasks, including computer vision (classification, detection, and segmentation), natural language processing (word analogy, and text classification), and fine-tuning settings on large backbones.

## 1. Introduction

Disentanglement learning is widely regarded as a cornerstone for achieving model generalization in deep learning (Bengio et al., 2013; Wang et al., 2024). The fundamental objective is to identify and separate the underlying Factors of Variation (FoV) governing the data generation process (Kingma & Welling, 2013; Higgins et al., 2017; Chen et al., 2018; Kim & Mnih, 2018; Zhu et al., 2021; Jung et al., 2024; Yang et al., 2022; Keller & Welling, 2021a; Jeong & Song, 2019; Keller & Welling, 2021b; Shao et al., 2020; 2022; Jung et al., 2026). This pursuit is motivated by the premise that a representation reflecting the true compositional structure of the data is expected to facilitate generalization to unseen scenarios (Bengio et al., 2013; Wang et al., 2024; Locatello et al., 2019a). Consequently, the prevailing literature predominantly focuses on constructing a latent vector space where these factors are encoded in a dimension-wise or vector-wise manner (Wang et al., 2024).

These latent-vector-centric methodologies typically assume that observations are generated from (approximately) independent factors of variation (FoV), and inject inductive bias to separate them from two complementary perspectives: statistical and geometric constraints on the latent vector space. Statistical approaches promote a factorial latent space by penalizing dependence across latent dimensions (Higgins et al., 2017; Chen et al., 2018; Kim & Mnih, 2018; Shao et al., 2020; Jeong & Song, 2019), while geometric approaches incorporate symmetry constraints (e.g., group-based equivariance) to align latents with structured transformations of FoV (Zhu et al., 2021; Yang et al., 2022; Jung et al., 2024). However, prior analyses indicate that optimizing latent objectives alone does not reliably guarantee disentanglement quality (Zietlow et al., 2021), and improvements in intrinsic disentanglement metrics do not consistently translate into downstream performance gains (Locatello et al., 2019b; 2020a). Since downstream predictions are ultimately mediated by intermediate features throughout the network (Yosinski et al., 2014; Zeiler & Fergus, 2013; Bengio et al., 2013), these limitations motivate moving beyond latent-vector-centric constraints toward architectural inductive biases that make the network structurally better suited to disentangle FoV and downstream generalization.

Complementary to latent-vector-centric methods, architectural approaches that explicitly disentangle semantic or structural components of the input provide a compelling alternative. Prior works introduce specialized modules that explicitly separate geometric structures (e.g., walls vs. ceilings), leading to improved detection and segmentation performance in computer vision (Yin et al., 2020; Gkitsas et al., 2021). Similarly, architectures that disentangle objects from attributes can yield substantial gains in compositional zero-shot learning (Saini et al., 2022). Also, (Hu et al., 2021) disentangle pre-trained model weight into task-specific sub-architectures to analyze its internal inference process. Differently, we propose the Orthogonal Subspaces Projection

[1]Department of AI Convergence, Gwangju Institute of Science and Technology (GIST), South Korea. Correspondence to: Kangil Kim <kangil.kim.01@gmail.com>.

*Proceedings of the 43rd International Conference on Machine Learning*, Seoul, South Korea. PMLR 306, 2026. Copyright 2026 by the author(s).

(OSP) layer, a plug-and-play architectural module that enforces separability by projecting arbitrary intermediate activations into orthogonal subspaces differen. By shifting part of the disentanglement inductive bias from latent-space objectives to a reusable layer design, OSP is naturally compatible with large-scale backbones and supports systematic evaluation across diverse downstream tasks.

Specifically, the OSP layer is configured to project latent features into $K$ mutually orthogonal subspaces, thereby forming an independent subspace structure that mirrors the common assumption of (approximately) independent FoV. This layer generates a new output feature vector through a two-stage process: (1) mapping input features into orthogonal subspaces to suppress cross-subspace interference, and (2) dimensionality compression that retains dominant directions to reduce redundancy. Crucially, this architecture admits an efficient implementation with lightweight computational overhead. Unlike conventional latent-vector-centric methods, the OSP layer acts as a modular component that can be integrated atop any standard layer (e.g., convolutional or linear layers) within existing architectures. This design enables the model to learn disentangled representations progressively through continuous layer-centric operations, rather than enforcing constraints solely at the latent vector space with an objective function.

Our contributions are summarized as follows:

- **Layer-Centric Disentanglement Learning**: To the best of our knowledge, we are the first to propose a continuous layer-centric method that improves disentanglement quality as an alternative to latent-vector-centric approaches.

- **Model Agnostic Plug-and-Play Layer**: The OSP layer can be applied to any architecture that consists of convolutional and/or linear layers, including large models such as ResNet (He et al., 2016), Transformer-based architectures (Dosovitskiy et al., 2020; Devlin et al., 2019; Hu et al., 2022).

- **Extended Validation on Diverse Downstream Tasks and Enhanced Generalization**: Models equipped with the OSP layer improve downstream generalization within Computer Vision (CV), Natural Language Processing (NLP), and Fine-Tuning tasks (CV classification).

## 2. Related Works

### 2.1. Latent-Vector-Centric Disentanglement Learning

Unsupervised latent-vector-centric approaches seek to align distinct latent coordinates with factors of variation (FoV) by promoting an approximately factorial latent structure.

As (Wang et al., 2024) investigates and clarifies the dimension-wise disentangled representation, we follow the definition. Accordingly, representative methods encourage independence by penalizing statistical dependence in the latent space using measures such as mutual information or total correlation (Higgins et al., 2017; Chen et al., 2018; Kim & Mnih, 2018; Shao et al., 2020; 2022). Complementary to purely unsupervised formulations, weakly/semi-supervised approaches suggest that limited supervision can be necessary to more reliably align latent dimensions with semantic factors (Trauble et al., 2020; Locatello et al., 2020b; Hajimiri et al., 2021). Another line of work incorporates group-theoretic inductive bias to encode known symmetries of factors, e.g., by imposing equivariant constraints or structured latent objectives in VAE-based frameworks (Painter et al., 2020; Zhu et al., 2021; Yang et al., 2022; Jung et al., 2024; Keurti et al., 2023; Hsu et al., 2023). While prior methods have improved disentanglement learning on controlled benchmarks (Reed et al., 2015; Matthey et al., 2017; Burgess & Kim, 2018; Gondal et al., 2019), evidence that these intrinsic gains reliably translate into tangible improvements in downstream generalization remains limited (Locatello et al., 2019b; 2020a). In contrast, we propose an approach that explicitly targets broad downstream generalization and is realized as a plug-and-play structural module compatible with large-scale models.

### 2.2. Orthogonality-based layers and subspace methods

Orthogonality has been exploited through several design axes, depending on where it is enforced. A first line focuses on feature/head-level designs for recognition. CapProNet projects learned features onto capsule subspaces via orthogonal projections for classification (Zhang et al., 2018), and LSOP learns subspace orthogonal projections tailored to semi-supervised image classification (Li et al., 2022). Relatedly, orthogonality can be embedded into the classification layer itself (Saadeldin & Mac Namee, 2021). A second line enforces orthogonality at the objective level, e.g., by adding an auxiliary loss that encourages orthogonal separation in the feature space (Ranasinghe et al., 2021). While effective in their respective settings, these approaches are typically tied to specific components (e.g., heads or losses), rather than providing a general-purpose module that can be inserted to project arbitrary intermediate activations into mutually orthogonal subspaces across architectures.

More recently, orthogonality has also been leveraged from a weight-centric perspective for training and adaptation. Orthogonal over-parameterized training (OPT) formulates orthogonality primarily as a training reparameterization that learns orthogonal transforms over fixed base weights to preserve favorable neuron geometry during optimization (Liu et al., 2020). Orthogonal fine-tuning (OFT) further adapts large pre-trained backbones by learning multiplicative or-

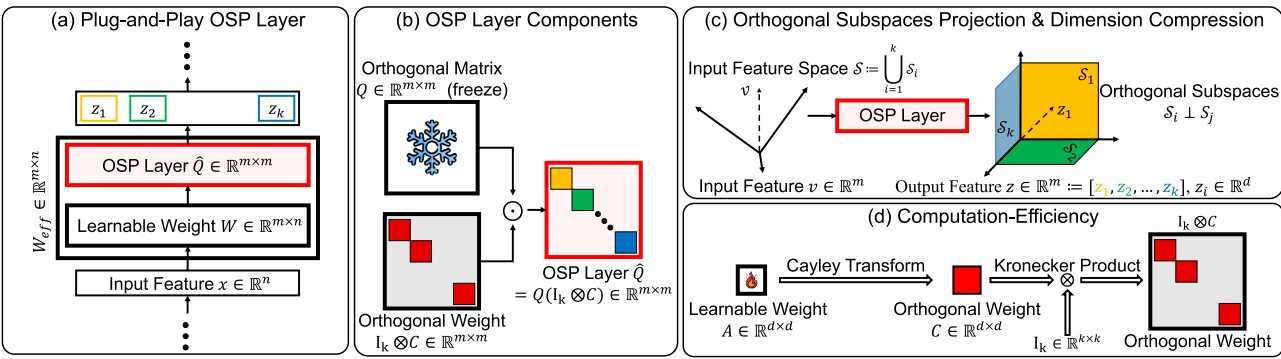

*Figure 1.* Overview: Orthogonal Subspaces Projection (OSP) Layer. (a) shows how OSP layer is applied to a linear layer including CNN layer, (b) shows the components of OSP layer which consist of non-trainable orthogonal matrix $Q$, and learnable block diagonal matrix $I_k \otimes C$, (c) visualize the effects of OSP that project to orthogonal subspaces and dimension compression to remove redundant dimension, and (d) shows the computational efficiency of OSP $\mathcal{O}(d^3)$ instead of $\mathcal{O}(m^3)$, where $d << m$.

thogonal transforms on bottom of frozen weights (Qiu et al., 2023), BOFT improves parameter efficiency via butterfly-structured orthogonal parameterizations (Liu et al., 2023), and subsequent work addressing scalability of such orthogonal adapters (Qiu et al., 2025). Differently our OSP layer projects intermediate features into $K$ mutually orthogonal subspaces on top of any learnable/frozen weights. This distinguishes OSP from head/loss-specific formulations and from weight-reparameterization/adaptation schemes, and enables a unified evaluation across diverse architectures and downstream tasks.

## 3. Methods

As shown in Figure 1, the proposed orthogonal subspaces projection (OSP) layer can be inserted on top of any standard linear layers ($W$) at any intermediate depth, including convolutional layers. In this section, we present the OSP layer by (1) constructing orthogonal projection matrices that map output features onto mutually orthogonal subspaces to encourage disentangled representations, (2) performing dimensionality compression to retain the principal components for efficiency of computational complexity and memory usage, and (3) introducing an efficient tensorized implementation to reduce computational overhead.

### 3.1. Orthogonal Subspaces Projection for Layer-Centric Disentanglement Learning

**Motivation** Intermediate features play a central role in shaping downstream predictions and generalization (Yosinski et al., 2014; Zeiler & Fergus, 2013; Bengio et al., 2013). In contrast, many dimension-wise disentanglement methods impose regularizers primarily on latent variables to encourage the separation of factors of variation (Wang et al., 2024), yet evidence that such latent-space objectives reliably translate into improved downstream generalization remains limited (Locatello et al., 2019b; 2020a). To address this gap, we adopt an architectural approach that promotes layer-centric

disentanglement by projecting intermediate activations onto mutually orthogonal subspaces. This complements latent-vector-centric approaches by injecting disentangling bias directly into the network's intermediate representations rather than only through a latent-space objective.

**Orthogonal Subspaces Projection** Motivated by disentanglement learning methods that conduct objectives on each dimension of latent vectors to be independent, we enforce divided subspaces to be orthogonal to encourage independent effect of disentanglement learning methods. Let $A' \in \mathbb{R}^{m \times m}$, and consider its QR decomposition $A' = QR$, where $Q \in \mathbb{R}^{m \times m}$ has orthonormal columns (i.e., $Q^\top Q = I_m$) and $R \in \mathbb{R}^{m \times m}$ is upper triangular. We further partition $Q$ into $k$ column blocks as $Q = [Q_1, Q_2, \ldots, Q_k]$, where each $Q_i \in \mathbb{R}^{m \times d}$ and $m = d \cdot k$. Then the matrix $P_i := Q_i Q_i^\top \in \mathbb{R}^{m \times m}$ is the orthogonal projection onto the $d$-dimensional subspace $S_i := \mathrm{col}(Q_i) \subset \mathbb{R}^m$ spanned by the columns of $Q_i$, where $S_i \perp S_j$. Accordingly, for any $v \in \mathbb{R}^m$, $P_i v = Q_i Q_i^\top v$ is the orthogonal projection of $v$ onto $S_i$, where $P_i v \in \mathbb{R}^m$, then any projection vectors are orthogonal ($P_i v \perp P_j v$).

**Dimension Compression via Intrinsic Subspace Coordinates** Naively generating the set of $k$ projected vectors $\{P_i v\}_{i=1}^k$ would incur a prohibitive increase in both computational complexity and memory usage (specifically, a $k$-fold expansion). However, this overhead stems largely from redundant dimensions: although each projected vector $P_i v$ resides in the $m$-dimensional ambient space, it is strictly confined to a lower $d$-dimensional subspace $S_i$ (i.e., $\mathrm{rank}(P_i) = d \ll m$). To eliminate this redundancy and reduce costs, we represent each projection using its intrinsic coordinates with respect to the orthonormal basis $Q_i$: $z_i := Q_i^\top v \in \mathbb{R}^d$. This representation is lossless regarding the projected information (since $P_i v = Q_i z_i$) yet significantly more compact. Finally, we concatenate these subspace coordinates to form the output $z := [z_1; \ldots; z_k] \in \mathbb{R}^{kd} = \mathbb{R}^m$, thereby avoiding the storage and computation explosion

while preserving the full information of the orthogonal projections.

**Computation-Efficient Implementation.** Directly optimizing the full orthogonal basis $Q \in \mathbb{R}^{m \times m}$ is computationally prohibitive, as maintaining orthogonality typically requires $\mathcal{O}(m^3)$ retractions (e.g., QR/SVD) at each update. To avoid this overhead, we reparameterizes the basis as a composition of a fixed global reference frame and a low-dimensional learnable rotation. Specifically, we store a fixed orthogonal matrix $Q_{\text{total}}$ as a non-trainable buffer (initialized once, e.g., by a QR decomposition, or set to $I$ for pretrained initialization), and learn only a shared subspace rotation $C \in \mathbb{R}^{d \times d}$ applied to every block of columns. Let $Q_{\text{total}} = [Q_1, \ldots, Q_k]$ with $Q_i \in \mathbb{R}^{m \times d}$ and $m = kd$; we form the effective basis by $\hat{Q} = [Q_1 C, \ldots, Q_k C] = Q_{\text{total}}(I_k \otimes C)$, which preserves orthonormality within each block since $Q_i$ has orthonormal columns and $C$ is orthogonal. To enforce $C^\top C = I_d$ without expensive manifold constraints, we parameterize $C$ via the Cayley transform ([Helfrich et al.](), [2017]()) using a learnable skew-symmetric matrix $A^\top = -A$:

$$C = (I_d - A)(I_d + A)^{-1}, \tag{1}$$

so the additional cubic cost is confined to inverting a $d \times d$ system, i.e., $\mathcal{O}(d^3)$ with $d \ll m$. In implementation, $Q_{\text{total}}$ is reshaped into $k$ blocks and right-multiplied by $C$ via batched matrix multiplication, avoiding explicit construction of the projectors $\{P_i\}$.

**Orthogonal Subspaces Projection (OSP) Layer** As shown in Figure 1 (b), OSP layer is formulated as:

$$OSP(v) = \hat{Q}v = Q(I_k \otimes C)v, \tag{2}$$

where $\otimes$ is a Kronecker product.

### 3.2. Plugging onto Linear and Convolutional Layers

The proposed OSP can be used as a drop-in replacement for standard learnable layers, including linear and convolutional layers.

**Linear layer (OSP-Linear)** Given a standard linear map $y = Wx + b$ with $W \in \mathbb{R}^{m \times n}$, we define the OSP-parameterized weight as

$$W_{\text{eff}} := \hat{Q}W, \qquad \text{and compute} \qquad y = W_{\text{eff}}x + b. \tag{3}$$

In code, $\hat{Q}$ is constructed from the fixed buffer $Q_{\text{total}}$ by reshaping it into $k$ blocks and right-multiplying each block by the same $C \in \mathbb{R}^{d \times d}$, where $C$ is obtained by a Cayley transform of a skew-symmetric matrix derived from a learnable parameter. For fine-tuning compatibility, the pretrained variant initializes $Q_{\text{total}} = I$ and $C = I$, so the layer starts exactly as a standard linear layer.

**Convolutional layer (OSP-Conv)** For a convolution with kernel $W \in \mathbb{R}^{m \times c_{\text{in}} \times k_h \times k_w}$ (with $m = c_{\text{out}}$), we apply the same reparameterization on the output-channel dimension by flattening the kernel to $W_{\text{flat}} \in \mathbb{R}^{m \times (c_{\text{in}} k_h k_w)}$ and forming

$$\begin{aligned} W_{\text{flat,eff}} &:= \hat{Q}\,W_{\text{flat}}, \\ W_{\text{eff}} &:= \text{reshape}\Big(W_{\text{flat,eff}},\ (m,\ c_{\text{in}},\ k_h,\ k_w)\Big). \end{aligned} \tag{4}$$

The forward pass then uses $W_{\text{eff}}$ in a standard `conv2d` call, so OSP-Conv is plug-and-play wherever a conventional convolution is used. The shared within-subspace rotation $C$ is computed via the Cayley transform by solving a $d \times d$ linear system (equivalently multiplying by $(I + A)^{-1}$), ensuring $C$ is orthogonal whenever $(I + A)$ is invertible.

## 4. Experiments

We adopt a two-stage evaluation protocol. First, we train models on disentanglement learning benchmarks to quantitatively assess whether the equipped OSP layer model separates Factors of Variation (FoV) and to analyze the resulting representations. Second, to validate practical utility, we conduct comprehensive downstream experiments spanning computer vision (classification, detection, and segmentation), natural language processing (word analogy and text classification), and fine-tuning settings with large-scale models.

### 4.1. Disentanglement Learning

**Datasets** We utilize the 3D Shapes ([Burgess & Kim](), [2018]()), and MPI3D ([Gondal et al.](), [2019]()) datasets for disentanglement learning tasks. The 3D Shapes dataset consists of 480,000 RGB $64 \times 64 \times 3$ images with six independent ground truth factors: orientation (15), shape (4), floor color (10), scale (8), object color (10), and wall color (10) ([Burgess & Kim](), [2018]()). More details are in Appendix A.1

**Hyper-Parameter Tuning** We implement $\beta$-VAE ([Higgins et al.](), [2017]()), $\beta$-TCVAE ([Chen et al.](), [2018]()), and Commutative Lie Group VAE (CLG-VAE) ([Zhu et al.](), [2021]()) as baselines. For common settings to baselines, we set batch size 1024, iteration 31250, and random seed from $\{1, 2, \ldots, 10\}$ without weight decay, and learning rate $\in \{16\text{e-}4, 1\text{e-}4\}$ as respect to 3D Shapes and MPI3D on a single RTX 2080 Ti GPU. We equipped OSP layers on only encoder part. More details are in Appendix A.1.

**Disentanglement Metrics** To estimate the FVM score proposed in ([Kim & Mnih](), [2018]()), we compute the global empirical variance across each dimension using a sample size of 100, repeating the process for 800 iterations. For the beta-VAE ([Higgins et al.](), [2017]()), MIG ([Chen et al.](), [2018]()),

*Table 1.* Disentanglement performance of AE-based and VAE-based models on 3D Shapes and MPI3D datasets. Results are reported as mean ± std over three seeds. Bold text indicates improved scores over the corresponding baseline model.

*(a)* AE vs. OSP-AE.

| Dataset | Metric | AE | OSP-AE |
|---|---|---|---|
| 3D Shapes | $\beta$-VAE | 67.20(±2.28) | **76.80**(±5.02) |
| | FVM | 50.62(±4.15) | **60.73**(±7.00) |
| | MIG | 5.46(±1.18) | **8.87**(±1.58) |
| | SAP | 2.23(±0.23) | **2.93**(±0.49) |
| | DCI-Dis. | 10.52(±2.09) | **14.66**(±2.69) |
| | DCI-Com. | 11.34(±1.50) | **16.26**(±1.98) |
| MPI3D-toy | $\beta$-VAE | 55.20(±5.22) | **61.33**(±4.62) |
| | FVM | 33.48(±2.85) | **37.67**(±4.57) |
| | MIG | 1.99(±0.38) | **2.65**(±1.61) |
| | SAP | 1.23(±0.38) | **1.40**(±0.33) |
| | DCI-Dis. | 7.70(±1.00) | **8.59**(±2.27) |
| | DCI-Com. | 7.53(±1.03) | **8.18**(±2.11) |

*(b)* Disentanglement performance on 3D Shapes and MPI3D datasets.

| Dataset | Model | $\beta$-VAE | FVM | MIG | SAP | DCI-disen. | DCI-compl. |
|---|---|---|---|---|---|---|---|
| 3D Shapes | $\beta$-VAE | 85.80(±8.66) | 75.86(±10.73) | 35.41(±17.06) | 6.90(±2.61) | 46.31(±16.36) | 48.28(±16.68) |
| | OSP-$\beta$-VAE | **90.60**(±4.72) | **78.28**(±10.06) | **43.33**(±13.95) | **7.50**(±1.43) | **53.68**(±12.83) | **56.69**(±11.65) |
| | $\beta$-TCVAE | 88.80(±5.02) | 74.70(±3.39) | 34.27(±12.54) | 6.51(±1.46) | 46.62(±14.28) | 48.57(±13.56) |
| | OSP-$\beta$-TCVAE | **90.80**(±5.22) | **84.88**(±4.64) | **41.32**(±14.06) | **8.48**(±2.44) | **54.44**(±9.92) | **55.85**(±10.14) |
| | CLG-VAE | 79.60(±18.83) | 68.80(±18.13) | 34.41(±14.13) | 8.80(±4.25) | 40.51(±15.16) | 52.73(±19.11) |
| | OSP-CLG-VAE | **85.80**(±8.30) | **78.68**(±10.59) | **34.79**(±12.42) | **9.05**(±2.56) | **42.53**(±11.42) | **53.51**(±10.27) |
| MPI3D | $\beta$-VAE | 37.20(±16.39) | 33.34(±10.90) | **6.91**(±9.78) | 3.73(±4.33) | 13.48(±9.93) | 21.09(±14.93) |
| | OSP-$\beta$-VAE | **43.80**(±12.73) | **38.20**(±8.69) | 5.83(±6.35) | **4.05**(±3.33) | **15.26**(±6.53) | **25.01**(±9.34) |
| | $\beta$-TCVAE | 34.40(±15.19) | 32.00(±12.35) | 6.42(±11.09) | 3.94(±5.34) | 11.93(±11.75) | 18.96(±17.70) |
| | OSP-$\beta$-TCVAE | **46.00**(±6.16) | **37.53**(±4.37) | **17.36**(±7.73) | **9.50**(±3.60) | **15.39**(±1.60) | **26.53**(±0.79) |
| | CLG-VAE | 41.00(±5.03) | 34.97(±1.04) | 19.30(±1.78) | 10.22(±0.74) | 19.17(±2.42) | **29.64**(±1.45) |
| | OSP-CLG-VAE | **48.00**(±3.65) | **39.86**(±1.39) | **19.90**(±1.09) | **10.40**(±0.33) | 19.05(±2.12) | 29.41(±1.24) |

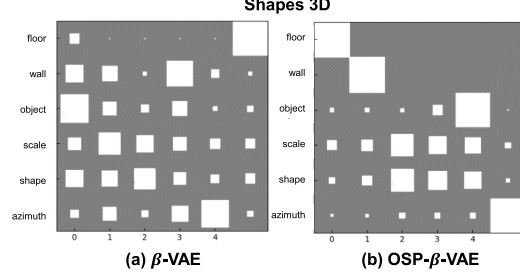

(a) $\beta$-VAE     (b) OSP-$\beta$-VAE

*Figure 2.* We visualize the DCI matrix to demonstrate the alignment between latent codes and ground-truth factors. Here, $r_{k,j}$ denotes the importance of the latent unit $z_j$ in predicting the factor $v_k$, covering the 1) six factors of the 3D Shapes dataset (i.e., $v_k \in \{\text{Floor, Wall, Object, Scale, Shape, Azimuth}\}$). Ideally, a disentangled representation manifests as a sparse matrix populated by isolated cells with high magnitude, indicating a one-to-one mapping between latents and factors.

SAP (Kumar et al., 2018), and DCI (Eastwood & Williams, 2018) metrics, we adhere to the standard protocols outlined in (Michlo, 2021), utilizing 100 mini-batches with 100 training rounds and 50 evaluation rounds, respectively. By employing this set of four distinct metrics, we aim to provide a comprehensive and less biased assessment of the disentanglement performance.

**Pure Structural Impact and Quantitative Results** To isolate the structural effect of OSP, we compare AE and OSP-AE instead of VAE-based models with explicit disentanglement objectives. This setting removes the influence of KL divergence and other disentanglement-specific regularizations used in prior methods. As shown in Table 1a, OSP-AE consistently outperforms AE, indicating that the OSP structure itself improves disentanglement learning. As shown in Table 1b, applying the OSP layer to CNN and linear layers outperforms standard VAE-based methods.

**Qualitative Results** As shown in Figure 2, DCI matrix (Eastwood & Williams, 2018) visualization of $\beta$-VAE and $\beta$-VAE with OSP layers on the 3D Shapes and MPI3D dataset, and our model is close to the ideal case (sparse matrix). It implies that our model contains information about a

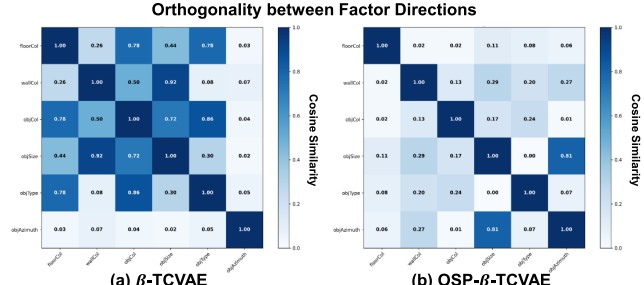

(a) $\beta$-TCVAE     (b) OSP-$\beta$-TCVAE

*Figure 3.* FoV direction orthogonality visualization.

single factor on a single latent vector dimension.

**Analysis: Does OSP disentangle FoV in the latent space?** We probe whether applying Orthogonal Subspaces Projection (OSP) encourages factor-wise disentanglement *geometrically* in the latent space by measuring the orthogonality among *factor direction vectors*. Let $\mu(x) \in \mathbb{R}^d$ denote the latent mean representation of an input $x$ (e.g., the encoder mean in a VAE). Given an "original" set $\mathcal{X}_{\text{orig}} = \{x_k^{\text{orig}}\}_{k=1}^N$ and a factor-specific set $\mathcal{X}_i = \{x_k^{(i)}\}_{k=1}^{N_i}$ where only factor $i$ is varied, we compute the original centroid

$$c_{\text{orig}} = \frac{1}{N} \sum_{k=1}^N \mu(x_k^{\text{orig}}), \qquad (5)$$

and define the centroid direction for factor $i$ as

$$v_i = c_i - c_{\text{orig}}, \qquad c_i = \frac{1}{N_i} \sum_{k=1}^{N_i} \mu(x_k^{(i)}). \qquad (6)$$

We then normalize each direction $\hat{v}_i = v_i / \max(\|v_i\|_2, \varepsilon)$ and form the absolute cosine similarity matrix

$$S_{ij} = |\hat{v}_i^\top \hat{v}_j| = \left| \frac{v_i^\top v_j}{\max(\|v_i\|_2, \varepsilon) \max(\|v_j\|_2, \varepsilon)} \right| \in [0, 1], \qquad (7)$$

where smaller off-diagonal values indicate more orthogonal (i.e., less coupled) factor directions, while $S_{ii} = 1$ holds by construction. We visualize $S$ as a heatmap with a fixed

*Table 2.* Computer vision classification task.

| Models | Params | CIFAR-100 | ImageNet |
|---|---|---|---|
| ResNet-18 | 11.69M | $72.97_{\pm1.38}$ | $64.44_{\pm0.37}$ |
| OSP-ResNet-18 | 11.70M | $\mathbf{74.00}_{\pm0.35}$ | $\mathbf{65.71}_{\pm0.23}$ |
| ResNet-50 | 25.56M | $73.28_{\pm0.61}$ | $69.86_{\pm0.12}$ |
| OSP-ResNet-50 | 25.58M | $\mathbf{74.64}_{\pm0.68}$ | $\mathbf{70.79}_{\pm0.14}$ |
| ViT-Base/16 | 87.40M | - | $78.22_{\pm0.45}$ |
| OSP-ViT-Base/16 | 87.58M | - | $\mathbf{80.33}_{\pm0.13}$ |

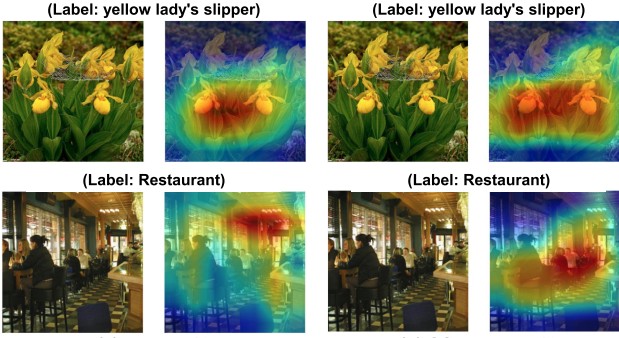

(a) ResNet 18    (b) OSP-ResNet 18

*Figure 4.* Lowest cross entropy loss value samples on each class.

range $[0, 1]$, where darker colors correspond to similarity value 1.

Figure 3 compares $\beta$-TCVAE and OSP-$\beta$-TCVAE. The OSP-equipped model exhibits systematically lower off-diagonal similarities, indicating that varying one factor shifts the latent representation along a direction that is closer to orthogonal to the directions induced by other factors. These results suggest that OSP promotes a more factor-separable organization of the latent space, consistent with improved disentanglement of FoV beyond the baseline.

### 4.2. CV: Image Classification

**Experimental Settings**    We evaluate computer-vision classification on CIFAR-100 (Krizhevsky, 2009) and ImageNet (Deng et al., 2009). As backbones, we use ResNet-18/50 (He et al., 2016) and a Vision Transformer base with $16 \times 16$ patches (ViT-B/16) (Dosovitskiy et al., 2020). To reduce the computing cost, we utilize the mixed precision training (Micikevicius et al., 2017). For CIFAR-100, we train with batch size 128 for 200 epochs using stocastic gradient decent (SGD) with learning rate $10^{-1}$ and weight decay $5 \times 10^{-4}$. For ImageNet with ResNet-18/50, we use Adam (Kingma & Ba, 2014) with learning rate $10^{-3}$, no weight decay, batch size 1024, and 100 epochs on 4 RTX 3090 GPUs. For ViT-B/16, we follow the training recipe in (Touvron et al., 2021) with additional data augmentations. We set $d = 32$ for OSP-ResNet-18/50, and $k = 256$ for OSP-ViT. More details are in Appendix A.2.

**Quantitative Results: Outperformance and Efficiency**
As shown in Table 2, inserting the OSP layer into every convolutional and linear layer consistently improves perfor-

mance over the original architectures on both CIFAR-100 and ImageNet across ResNet-18/50 and ViT-B/16. These results support the claim that OSP is model-agnostic, providing reliable gains when applied broadly within diverse backbones. Importantly, the parameter overhead is negligible (at most $\sim 0.2\%$), while top-1 accuracy increases by roughly 0.8-1.3 points in most settings, indicating that OSP offers an efficient accuracy-capacity trade-off.

**Impact on decision patterns (Grad-CAM)**    To examine how OSP layers influences classification decisions, we visualize the model's spatial evidence using Grad-CAM (Gradient-weighted Class Activation Mapping) (Selvaraju et al., 2017), which highlights the image regions most responsible for the predicted score. As shown in Figure 4, for each class label we select the sample that attains the lowest cross-entropy loss under the ground-truth label (i.e., the most confidently correct example for that class), and compare the corresponding Grad-CAM heatmaps between ResNet-18 and OSP-ResNet-18. Across the selected examples, OSP-ResNet-18 consistently places higher emphasis on semantically relevant object regions (e.g., the primary foreground instance) while reducing spurious attention to less informative background cues, in contrast to the baseline ResNet-18.

**Analysis on Hyper-spherical Energy Perspective**    Hyperspherical energy (HE) quantifies the diversity of neuron weight directions by penalizing small pairwise separations on the unit sphere (Liu et al., 2018; 2020). For each CNN layer $\ell$, let $\{\boldsymbol{w}_i^{(\ell)}\}_{i=1}^{N_\ell}$ denote the set of neuron

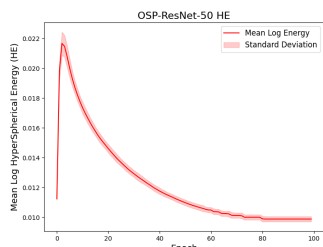

*Figure 5.* CNN layer-averaged relative log-energy $\overline{HE}$ of OSP-ResNet-50 during ImageNet dataset training

weight vectors in that layer (e.g., output channels/filters for convolution), and define $\hat{\boldsymbol{w}}_i^{(\ell)} := \boldsymbol{w}_i^{(\ell)}/\|\boldsymbol{w}_i^{(\ell)}\|$. With $s = 1$, we compute the layer-wise energy

$$E_1^{(\ell)} := \sum_{i=1}^{N_\ell} \sum_{j=1, j\neq i}^{N_\ell} \|\hat{\boldsymbol{w}}_i^{(\ell)} - \hat{\boldsymbol{w}}_j^{(\ell)}\|^{-1}, \quad (8)$$

and the corresponding simplex reference

$$\hat{E}_1^{(\ell)} = N_\ell(N_\ell - 1)\left(\frac{2N_\ell}{N_\ell-1}\right)^{-1/2}. \quad (9)$$

We then report the CNN layer-averaged relative log-energy

$$\overline{\mathrm{HE}} := \frac{1}{L}\sum_{\ell=1}^{L} \log\left(\frac{E_1^{(\ell)}}{\hat{E}_1^{(\ell)}}\right), \quad (10)$$

*Table 3.* Computer vision object detection and instance segmentation task.

| Dataset | Model | Box | | | | Mask | | | |
|---|---|---|---|---|---|---|---|---|---|
| | | Precision | Recall | mAP50 | mAP50-95 | Precision | Recall | mAP50 | mAP50-95 |
| CarParts | YOLOv8-large-seg | 57.37(±1.38) | 77.63(±3.36) | 67.77(±2.55) | 58.03(±1.37) | 57.93(±1.76) | 78.97(±2.97) | 69.13(±2.84) | 56.57(±1.59) |
| | OSP-YOLOv8-large-seg | **58.13**(±1.02) | **79.37**(±1.02) | **69.07**(±0.84) | **58.07**(±0.06) | **59.03**(±1.17) | **80.47**(±0.80) | **70.63**(±0.96) | **56.73**(±0.35) |
| MSCOCO | YOLOv8-large-seg | 71.43(±0.86) | 61.74(±1.04) | 67.49(±0.03) | 51.28(±0.05) | 71.30(±0.13) | 60.21(±0.49) | 64.93(±0.07) | 42.05(±0.00) |
| | OSP-YOLOv8-large-seg | **71.93**(±1.24) | **62.45**(±0.76) | **67.72**(±0.07) | **51.43**(±0.08) | **71.71**(±1.52) | **60.60**(±0.76) | **65.15**(±0.13) | **42.14**(±0.03) |

so that values closer to 0 indicate weight directions closer to the simplex-like ideal on average across layers. As shown in

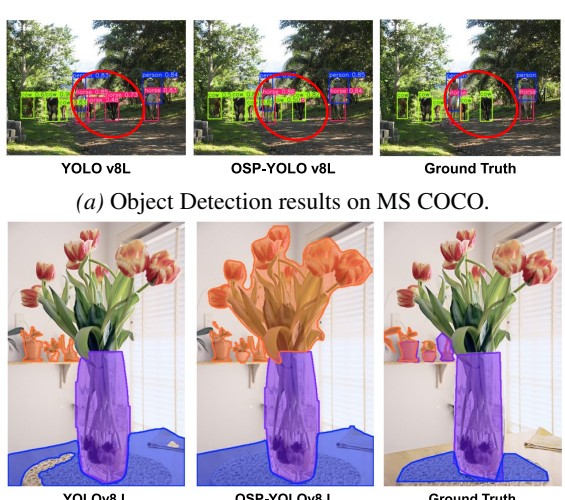

*(a)* Object Detection results on MS COCO.

*(b)* Instance Segmentation results on MS COCO.

*Figure 6.* Qualitative comparison between baseline YOLOv8 and OSP-YOLOv8. More details are in Figure 7-8.

Figure 5, $\overline{\text{HE}}$ for OSP-ResNet-50 decreases rapidly after the early training phase and stabilizes near zero, suggesting that OSP encourages increasingly well-spread weight directions throughout the network. This trend is consistent with prior observations that lower hyperspherical energy correlates with improved classification generalization (Liu et al., 2018; 2020).

### 4.3. CV: Object Detection and Instance Segmentation

**Experimental Settings** We employ YOLOv8 (Yaseen, 2024) as the detection baseline, specifically utilizing the 'Large' scale variants equipped with a segmentation head. The model is trained on CarParts (Pasupa et al., 2021), an automotive components dataset, and MS COCO (Lin et al., 2014) for general object detection. We resize all input images to $640 \times 640$ pixels and enable automatic mixed precision (AMP) to improve computational efficiency. We then train the models for 200 epochs on CarParts using AdamW (Loshchilov & Hutter, 2017) (momentum 0.9) with an initial learning rate of $3.7 \times 10^{-4}$ and a batch size of 32. For MS COCO (Lin et al., 2014), we train the models for 300 epochs using SGD (Bottou, 2010) (momentum 0.9) with an initial learning rate of $10^{-2}$ and a batch size of 128. We set $d = 8$ for OSP-YOLOv8. More details are in

Appendix A.3.

**Quantitative Results** Table 3 presents a comparative analysis between the baseline YOLOv8-large-seg and the proposed OSP-YOLOv8-large-seg on the CarParts and MS COCO datasets. The results demonstrate that the integration of the OSP layer consistently yields performance gains across both object detection (Box) and instance segmentation (Mask) tasks. Notably, the proposed model achieves superior performance across all evaluation metrics, including Precision, Recall, and mean average precision (mAP) at various thresholds, effectively attaining the best results (highlighted in bold) for both datasets. Such comprehensive improvements underscore the robustness of the OSP layer in enhancing the model's feature representation capabilities without introducing performance trade-offs.

**Qualitative Results** To further assess the practical effectiveness of the proposed method, we visualize qualitative detection and segmentation results in Figure 6. In the object detection task (Figure 6a), OSP-YOLOv8-large-seg demonstrates robust performance in cluttered and complex scenes. In the highlighted region, the baseline model misclassifies cow instances as horses. In contrast, the OSP-integrated model accurately identifies these instances as cows. Moreover, Figure 6b highlights a distinctive property of the proposed approach. OSP-YOLOv8-large-seg demonstrates robustness in segmenting semantically meaningful objects even when they are not annotated in the Ground Truth. This behavior suggests that the model exhibits improved generalization, extending beyond simple label fitting and capturing higher-level semantic characteristics of object structures.

### 4.4. NLP: Word Analogy

**Experimental Settings** We conducted a word analogy test, a common evaluation tool for measuring linguistic and cognitive abilities, using BERT (Devlin et al., 2018) and RoBERTa (Liu et al., 2019). The experimental setup was kept consistent with that (Ushio et al., 2021), while the OSP layers were added to all fully connected layers of each transformer layer for our models. The models were then trained on WikiText (Merity et al., 2016) for three epochs with a mini-batch size of 8, using masked language modeling with cross-entropy loss followed by a softmax function, optimized via SGD with a learning rate of 0.0001. The brief introduction of word analogy test and score functions for evaluating word analogies is provided in Appendix A.4.

*Table 4.* Word Analogy Test on various datasets. Best in bold. (The higher, the better.)

| | SAT | | | U2 | | | U4 | | | Google | | | BATS | | | Avg. | | |
|---|---|---|---|---|---|---|---|---|---|---|---|---|---|---|---|---|---|---|
| | ppl | mppl | pmi | ppl | mppl | pmi | ppl | mppl | pmi | ppl | mppl | pmi | ppl | mppl | pmi | ppl | mppl | pmi |
| BERT | 29.23 | **28.76** | 28.55 | 35.71 | 34.52 | **37.50** | **37.50** | 35.03 | **38.13** | 34.31 | 33.27 | 33.45 | 34.73 | **35.04** | 36.46 | 34.30 | 33.32 | **34.82** |
| OSP-BERT | **30.43** | 28.33 | **29.97** | 39.29 | 39.29 | 36.90 | 34.58 | **36.67** | 36.25 | 34.52 | 34.89 | 34.52 | 43.76 | 34.44 | 34.43 | **34.71** | **34.73** | 34.42 |
| RoBERTa | **30.51** | 29.03 | 29.86 | 36.76 | 34.33 | 35.71 | 34.43 | 35.21 | 35.83 | 33.06 | **33.41** | 32.83 | 33.25 | 33.25 | **33.87** | 33.60 | 33.04 | 33.86 |
| OSP-RoBERTa | 29.95 | **30.19** | **31.99** | 39.48 | 37.90 | 37.90 | 38.75 | 38.33 | 38.96 | 33.21 | 32.85 | **33.77** | 33.97 | 34.16 | 33.66 | **35.07** | **34.69** | 35.26 |

*Table 5.* GLUE task with BERT and RoBERTa models. (The higher, the better.)

| | Param. | WNLI | RTE | STS-B | MRPC | CoLA | SST-2 | QNLI | QQP | MNLI |
|---|---|---|---|---|---|---|---|---|---|---|
| BERT | 109.48M | 39.44 | 60.29 | 86.29/86.08 | 81.37/87.21 | **53.91** | 92.32 | 90.66 | 90.51/87.31 | 83.91/84.10 |
| OSP-BERT | 109.63M | **46.48** | **65.89** | **86.38/86.40** | **81.62/87.44** | 53.39 | **92.43** | **91.16** | **90.60/87.43** | **84.52/84.48** |
| BERT-Large | 335.14M | **56.33** | 63.54 | 87.38/87.49 | 81.62/87.09 | 57.82 | 92.78 | **92.26** | 91.31/88.40 | 86.34/86.44 |
| OSP-BERT-Large | 335.40M | **56.33** | **94.62** | **88.36/88.38** | **82.60/87.80** | **58.59** | **93.00** | 92.22 | **91.42/88.50** | **86.68/86.75** |
| RoBERTa-large | 355.43M | **56.34** | **72.20** | 91.09/91.17 | 87.99/91.51 | **64.07** | 95.99 | 94.33 | 91.75/89.09 | 90.74/90.48 |
| OSP-RoBERTa-large | 355.76M | **56.34** | **72.20** | **91.25/91.34** | **88.73/91.90** | 63.99 | **96.33** | **94.42** | **91.91/89.30** | **90.98/90.76** |

**Quantitative Results** The word analogy performance of BERT and RoBERTa, together with our proposed OSP layers replacement, is evaluated across multiple benchmark datasets (SAT, U2, U4, Google, and BATS) under three scoring metrics: perplexity-based (*ppl*), marginal likelihood–biased perplexity (*mppl*), and PMI-based (*pmi*). As shown in Table 4, overall, incorporating OSP layers consistently improves performance over the corresponding backbone models across all metrics and datasets. For BERT, OSP increases the average score from 34.30 to 34.71 under *ppl* and from 33.32 to 34.73 under *mppl*, while achieving competitive results from 34.82 to 34.42 under *pmi*. More pronounced improvements are observed for RoBERTa, where OSP yields consistent and often larger performance gains. In particular, OSP-RoBERTa outperforms the vanilla RoBERTa across almost all evaluated datasets and metrics, achieving the best overall average scores of 35.07 *ppl*, 34.69 *mppl*, and 35.26 *pmi*. These results empirically demonstrate that OSP layers effectively train the word relationship on the embedding space.

### 4.5. NLP: Text Classification (GLUE)

We conduct text classification experiments on the General Language Understanding Evaluation (GLUE) (Wang et al., 2018) benchmark with BERT and RoBERTa models.

**Experimental Settings.** We utilize pre-trained models (BERT-base, BERT-large, and RoBERTa) and fine-tune models on each dataset. We use 3 epochs for WNLI and MRPC (5 epochs), a learning rate of $2\times10^{-5}$, and a batch size of 32 with the AdamW optimizer. To apply OSP layers, we initialize the frozen weight $Q$ and the learnable weight $A$ as identity matrices.

**Results.** Table 5 reports the GLUE results of BERT- and RoBERTa-based models. Overall, applying the OSP layer consistently improves the performance of the corresponding backbone across most tasks while introducing only a

*Table 6.* Fine-tuning: computer vision classification task.

| Models | Fine-Tuning | # of Param. ↓ | ImageNet ↑ |
|---|---|---|---|
| | Full fine-tuning | 11.69M | 66.69(±0.06) |
| | OPT (Liu et al., 2020) | 46.50M | OOM |
| ResNet 18 | S-OPT (Liu et al., 2020) | 3.39M | 67.74 |
| | S-OPT | 1.80M | 50.07(±3.08) |
| | OSP-fine-tuning | 0.50M | **67.78**(±0.08) |
| ResNet 50 | Full fine-tuning | 25.56M | 76.59(±0.08) |
| | OSP-fine-tuning | 1.00M | **77.67**(±0.08) |

negligible increase in the number of parameters. In particular, OSP-BERT outperforms BERT on most reported tasks except CoLA, and OSP-BERT-Large also achieves improvements on nearly all tasks, with especially large gains on RTE. Similarly, OSP-RoBERTa-Large improves RoBERTa-Large on most tasks, including STS-B, MRPC, SST-2, QNLI, QQP, and MNLI, while remaining comparable on WNLI and slightly lower on CoLA. These results suggest that OSP serves as an effective and lightweight plug-in module that consistently enhances language understanding performance across different backbone scales and architectures.

### 4.6. Fine-Tuning

**Experimental Settings** We evaluate parameter-efficient fine-tuning on ImageNet using ResNet-18/50 backbones. We define OSP-fine-tuning that freeze pre-trained weights and update only the additional OSP layers. To preserve the exact functionality of the pre-trained network at initialization, we initialize the orthogonal basis $Q$ and the within-subspace rotation $C$ as identity matrices. We compare against full fine-tuning and OPT/S-OPT (Liu et al., 2020), we use Adam (Kingma & Ba, 2014) with learning rate $10^{-3}$, no weight decay, batch size 1024, and 60 epochs on 4 RTX 3090 GPUs.

**Quantitative Results** Table 6 demonstrates that OSP attains the best accuracy with the fewest trainable parameters. On ResNet-18, OSP updates only 0.5M parameters yet improves ImageNet top-1 accuracy from 66.69±0.06 (full fine-tuning, 11.69M trainable parameters) to 67.78±0.08, yield-

*Table 7.* Hyper-parameter sensitivity.

*(a)* Model performance over subspace dimension size.

| sub.dim. | BERT-base | | | RoBERTa-large | | |
|---|---|---|---|---|---|---|
| | RTE | SST2 | QQP | RTE | SST2 | QQP |
| 16 | 62.09 | 91.86 | 90.41/87.18 | 69.31 | 96.22 | 91.88/89.23 |
| 32 | 61.37 | 91.97 | 90.49/81.28 | **72.56** | 96.22 | 91.85/89.18 |
| 64 | **65.89** | **92.43** | **90.60/87.43** | 72.20 | **96.33** | **91.91/89.30** |
| 128 | 57.76 | 91.97 | 90.56/87.36 | 61.01 | 96.22 | 91.87/89.25 |

*(b)* Model performance over the number of subspaces.

| num.sub. | BERT-base | | | RoBERTa-large | | |
|---|---|---|---|---|---|---|
| | RTE | SST2 | QQP | RTE | SST2 | QQP |
| 16 | 60.65 | 92.09 | **90.53/87.32** | **71.12** | **96.33** | 91.85/89.22 |
| 32 | 62.09 | 92.09 | 90.46/87.22 | 70.76 | 95.87 | 91.77/89.10 |
| 64 | **63.18** | 91.74 | 90.42/87.18 | 67.51 | 96.10 | **91.87/89.23** |
| 128 | 62.82 | **92.20** | 90.49/87.27 | 68.23 | 95.87 | 91.81/89.11 |

*Table 8.* Computational cost with 512 batch size.

| sub.dim. | Method | Training Time ↑ (batch/s) | Inference Time ↑ (batch/s) | VRAM ↓ (GB) |
|---|---|---|---|---|
| | ResNet50 | 2.36 | 8.87 | 3.97 |
| 32 | Full-QR-ResNet50 | 0.97 | 1.47 | 7.18 |
| | OSP-ResNet50 | 1.69 | 3.78 | 6.01 |
| 64 | Full-QR-ResNet50 | 0.87 | 1.38 | 6.92 |
| | OSP-ResNet50 | 1.34 | 3.27 | 6.00 |

ing a +1.09 gain while being substantially more parameter-efficient. OSP also matches or slightly surpasses S-OPT (67.74) while using fewer trainable parameters (0.5M vs. 3.39M). Notably, OPT runs out of memory (OOM) even on ResNet-18 in our setting, indicating unfavorable scaling. On ResNet-50, OSP again improves top-1 accuracy from $76.59 \pm 0.08$ (full fine-tuning, 25.56M trainable parameters) to $77.67 \pm 0.08$ while updating only 1.0M parameters ($\sim 25\times$ fewer trainable parameters), confirming that the proposed layer yields consistent gains with lightweight adaptation.

**Discussion: Why OSP-Fine-Tuning scales better than OPT in practice?** A key practical difference is how the added parameters scale with layer dimensionality. OPT-style orthogonal transformations are parameterized in the (flattened) input space, whose dimensionality grows with the effective input size of each layer (for convolution, this can scale with spatial resolution and channel dimensions), which can lead to large memory overhead. In contrast, OSP is parameterized in a low-dimensional subspace and its additional parameters scale primarily with the output-channel dimensionality, making it easier to integrate into larger backbones under the same training budget.

### 4.7. Additional Analysis

**Model Sensitivity over Hyper-Parameters.** Table 7 shows that OSP is reasonably stable across different hyper-parameter choices, while moderate configurations generally yield the best performance. For the subspace dimension, 64 gives the strongest overall results for BERT-base, achieving the best scores on RTE, SST-2, and QQP, whereas RoBERTa-large also performs best or near-best around 32-64. For the number of subspaces, the performance variation is relatively small across settings, indicating that OSP is not overly sensitive to this choice. Overall, these results suggest that OSP is robust to hyper-parameter changes, with the best trade-off

typically obtained at moderate subspace sizes and numbers of subspaces.

**Computing Cost.** Table 8 reports the computational cost with a batch size of 512. Compared with the baseline ResNet50, both Full-QR-ResNet50 and OSP-ResNet50 incur additional computational overhead, but OSP-ResNet50 is consistently more efficient than Full-QR-ResNet50 across all settings. For subspace dimension 32, OSP-ResNet50 achieves 1.69 batch/s in training and 3.78 batch/s in inference, compared with 0.97 and 1.47 for Full-QR-ResNet50, while also using less VRAM (6.01 GB vs. 7.18 GB). A similar trend is observed for subspace dimension 64, where OSP-ResNet50 remains faster (1.34 vs. 0.87 batch/s in training; 3.27 vs. 1.38 batch/s in inference) and more memory-efficient (6.00 GB vs. 6.92 GB). These results show that, although OSP introduces overhead relative to the vanilla backbone, it is substantially more efficient than the full QR-based alternative.

## 5. Conclusion

To address the limited evidence that latent-vector-centric disentanglement objectives reliably translate into downstream generalization, we proposed a layer-centric alternative: the Orthogonal Subspaces Projection (OSP) layer. OSP can be inserted on top of any learnable/frozen weights and promotes factorized intermediate representations by projecting activations into $K$ mutually orthogonal subspaces that mirror the common assumption of (approximately) independent factors of variation. By combining orthogonal subspaces projection with an efficient low-dimensional parameterization, OSP is practical to integrate into larger backbones. Empirically, we demonstrated that OSP provides consistent improvements across diverse architectures and downstream tasks, spanning CV (classification, detection, segmentation), NLP (word analogy and text classification), and parameter-efficient fine-tuning.

## Impact Statement

This work aims to advance machine learning by improving representation learning via a lightweight architectural module. Potential societal impacts depend on the application and deployment context; we do not anticipate ethical concerns beyond those generally associated with the development and use of general-purpose machine learning models.

### Acknowledgments

This work was supported by the National Research Foundation of Korea (NRF) grant funded by the Korea government (MSIT) (No.2022R1A2C2012054, Development of AI for Canonicalized Expression of Trained Hypotheses by Resolving Ambiguity in Various Relation Levels of Representation Learning, Contribution Rate: 90.0%), and Institute of Information & communications Technology Planning & Evaluation (IITP) grant funded by the Korea government (MSIT) (No.2019-0-01842, Artificial Intelligence Graduate School Program (GIST), Contribution Rate: 10.0%).

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

# A. Experimental Setting Details

## A.1. Disentanglement Learning

**Details of Datasets** We utilize the 3D Shapes (Burgess & Kim, 2018), and MPI3D (Gondal et al., 2019) datasets for compositional generalization and disentanglement learning tasks. The 3D Shapes dataset consists of 480,000 RGB $64 \times 64 \times 3$ images with six independent ground truth factors: orientation (15), shape (4), floor color (10), scale (8), object color (10), and wall color (10) (Burgess & Kim, 2018). The MPI3D (real-world complex) dataset consists of 460,800 RGB $64 \times 64 \times 3$ images with seven independent ground truth factors: color (4) shape (4), size (2), height (3), background color (3), horizontal (40), and vertical axis (40) (Gondal et al., 2019).

**Details of Experimental settings** We implement $\beta$-VAE (Higgins et al., 2017), $\beta$-TCVAE (Chen et al., 2018), and Commutative Lie Group VAE (CLG-VAE) (Zhu et al., 2021) as baselines. For common settings to baselines, we set batch size 1024, iteration 31250, and random seed from $\{1, 2, \ldots, 10\}$ without weight decay, and learning rate $\in \{16e\text{-}4, 1e\text{-}4\}$ with respect to 3D Shapes and MPI3D on a single RTX 2080 Ti GPU. To reduce the computing cost, we utilize the mixed precision training (Micikevicius et al., 2017). We set $\beta = 1$ for $\beta$-VAE (Higgins et al., 2017) and $\beta$-TCVAE (Chen et al., 2018), and we also follow the settings (Zhu et al., 2021) as $\lambda_{hessian} = 40.0$, $\lambda_{decomp} = 20.0$, $p = 0.2$, and balancing parameter of $loss_{\text{rec group}}$ as 0.2 for CLG-VAE. To equip the OSP layer, we set subspace dimension size $d = [16, 8, 8, 4]$ from bottom to top CNN layer, and set $d = [8, 1, 1]$ for linear layers of VAEs.

## A.2. CV: Classification

We evaluate computer-vision classification on CIFAR-100 (Krizhevsky, 2009) and ImageNet (Deng et al., 2009). As backbones, we use ResNet-18/50 (He et al., 2016) and a Vision Transformer base with $16 \times 16$ patches (ViT-B/16) (Dosovitskiy et al., 2020). To reduce the computing cost, we utilize the mixed precision training (Micikevicius et al., 2017). For CIFAR-100, we train with batch size 128 for 200 epochs using stocastic gradient decent (SGD) with learning rate $10^{-1}$ and weight decay $5 \times 10^{-4}$. For ImageNet with ResNet-18/50, we use Adam (Kingma & Ba, 2014) with learning rate $10^{-3}$, no weight decay, batch size 1024, and 100 epochs on 4 RTX 3090 GPUs. For ViT-B/16, we follow the training recipe in (Touvron et al., 2021): batch size 4096, 300 epochs, AdamW (Loshchilov & Hutter, 2017) with learning rate $10^{-3}$ and weight decay 0.3, a 5-epoch warmup, and standard data augmentations including RandAugment, Mixup, and CutMix on 8 RTX A6000 GPUs.

## A.3. CV: Object Detection and Segmentation

We employ YOLOv8 (Yaseen, 2024) as detection baselines, specifically utilizing the 'Large' scale variants equipped with segmentation head. The model is trained on CarParts (Pasupa et al., 2021), an automotive components dataset, and MS COCO (Lin et al., 2014) for general object detection. We resize all input images to $640 \times 640$ pixels and enable automatic mixed precision (AMP) to improve computational efficiency. For both datasets, we leverage the automatic optimizer selection feature (`optimizer=auto`) from the Ultralytics framework[1]. This mechanism applies differential weight decay strategies for various parameter groups—including weights, biases, and Batch Normalization layers—to preserve model expressivity. Specifically, we train the models for 200 epochs on CarParts using AdamW (momentum 0.9) with an initial learning rate of $3.7 \times 10^{-4}$ and a batch size of 32. For MS COCO, we train the models for 300 epochs using SGD (momentum 0.9) with an initial learning rate of $10^{-2}$ and a batch size of 128 on 4 RTX 3090 GPUs.

## A.4. NLP: Word Analogy Test

As introduced in (Ushio et al., 2021), word analogies have served as a standard intrinsic evaluation task for assessing the quality of word embeddings. Early work on Word2vec (Mikolov et al., 2013) demonstrated that embeddings could capture relational structure through simple linear operations in the vector space (e.g., $v_{\text{king}} - v_{\text{man}} + v_{\text{woman}} \approx v_{\text{queen}}$). The conceptual roots of this evaluation can be traced to connectionist theories in cognitive science (Feldman & Ballard, 1982), which postulated that neural networks can give rise to emergent concepts (Hopfield, 1982; Hinton, 1986) by learning distributed representations over a shared embedding space (Hinton, 1986). From this perspective, analogy tests remain an attractive tools for probing how well embeddings and language models encode abstract relational knowledge.

---

[1] https://github.com/ultralytics/ultralytics

**Task Description**   Given a query pair $(h_q, t_q)$ and a set of candidate pairs $\{(h_i, t_i)\}_{i=1}^n$, the objective is to identify the candidate whose relational structure is most similar to that of the query.

**Scoring Function.**   *Perplexity* (PPL) is commonly employed as a metric for sentence re-ranking. For a given sentence $\boldsymbol{x}$, perplexity can be computed for masked language models such as BERT and RoBERTa:

$$f(\boldsymbol{x}) = \exp\left(-\sum_{j=1}^m \log P_{\text{mask}}\left(x_j | \boldsymbol{x}_{\backslash j}\right)\right),$$

where $\boldsymbol{x}$ is tokenized as $[x_1, \ldots, x_m]$, $\boldsymbol{x}_{\backslash j} = [x_1, \ldots, x_{j-1}, \langle\text{mask}\rangle, x_{j+1}, \ldots, x_m]$, and $P_{\text{mask}}\left(x_j | \boldsymbol{x}_{\backslash j}\right)$ is the pseudo-likelihood of the masked token $x_j$.

As a result, PPL-inspired scoring function ($s_{PPL}$) is defined as

$$s_{PPL}(t_i, h_i | h_q, t_q) = -\frac{f(\mathcal{T}(h_q, t_q, h_i, t_i))}{\sum_{k=1}^n f(\mathcal{T}(h_q, t_q, h_k, t_k))},$$

where a prompting function $\mathcal{T}(w_1, w_2, w_3, w_4)$ converts input words into a sequence by a pre-defined template (e.g., $\mathcal{T}_{\text{to-as}}(w_1, w_2, w_3, w_4) =$ "$[w_1]$ is to $[w_2]$ as $[w_3]$ is to $[w_4]$").

Specifically, *marginal likelihood–biased perplexity* (mPPL) is also defined as

$$s_{mPPL}(t_i, h_i | h_q, t_q) = \log s_{PPL}(t_i, h_i | h_q, t_q) - \alpha_t \cdot \log P(t_i | h_q, t_q) - \alpha_h \cdot \log P(h_i | h_q, t_q),$$

where $\alpha_t$ and $\alpha_h$ are hyperparameters. The $s_{mPPL}$ augments the standard perplexity formulation by incorporating two bias terms. By adjusting $\alpha_t$ and $\alpha_h$, mPPL allows explicit control over the degree to which candidate answers containing words that are semantically close to the query pair are preferentially scored.

Although perplexity is effective at measuring sentence fluency, it may not be the best option for evaluating the plausibility of a given analogical proportion candidate. As an alternative, *point-wise mutual information* (PMI) has been explored, as it explicitly concentrates on the words involved in the two given pairs. Building on this idea, a PMI-inspired scoring function $s_{PMI}$ is defined as

$$s_{PMI}(t_i, h_i | h_q, t_q) = \mathcal{A}\left(r(t_i | h_i, h_q, t_q), r(h_i | t_i, h_q, t_q)\right),$$

where $r(t_i | h_i, h_q, t_q) = \log P(t_i | h_i, h_q, t_q) - \alpha \cdot \log P(t_i | h_q, t_1)$, $\alpha$ is a hyperparameter to control the effect of the marginal likelihood, and $\mathcal{A}$ is one of the basic aggregation operations such as `min`, `max`, and `mean`.

### A.5. Model-Agnostic Method

As the OSP layer improves model performance regardless of architectures and tasks, we employ the OSP layer to GAN- and Diffusion-based disentanglement learning methods, such as IB-GAN (Jeon et al., 2021), and DisDiff (Yang et al., 2023). As shown in Table 9, applying OSP outperforms the baselines across model architectures and datasets.

*Table 9.* Disentanglement performance with GAN- and Diffusion-based methods.

*(a)* DisDiff vs OSP-DisDiff

| 3D Shapes | FVM | MIG | DCI-disen. |
|---|---|---|---|
| DisDiff (Yang et al., 2023) | 90.20($\pm$4.30) | 56.71($\pm$3.30) | 72.30($\pm$1.30) |
| OSP-DisDiff | **92.00**($\pm$3.34) | **58.10**($\pm$3.17) | **75.36**($\pm$1.38) |

*(b)* IB-GAN vs OSP-IB-GAN

| CdSprites | FVM |
|---|---|
| IB-GAN | 75.10($\pm$5.16) |
| OSP-IB-GAN | **78.49**($\pm$3.65) |

## B. Object Detection and Segmentation Test

### B.1. Additional Quantitative Resutls

On YOLO11-large, OSP consistently improves the overall detection and segmentation performance on CarParts, while yielding smaller but mostly positive gains on MSCOCO. On CarParts, OSP-YOLO11 improves all box metrics over YOLO11-large-seg, increasing precision from 60.37 to 61.18, recall from 75.30 to 82.05, mAP50 from 68.00 to 68.95, and mAP50-95 from 58.00 to 59.64. In the mask setting, applying OSP layer also improves all metrics, with precision rising from

*Table 10.* Additional Computer vision object detection and instance segmentation task with YOLO-v11-large.

| Dataset | Model | Box | | | | Mask | | | |
|---|---|---|---|---|---|---|---|---|---|
| | | Precision | Recall | mAP50 | mAP50-95 | Precision | Recall | mAP50 | mAP50-95 |
| CarParts | YOLOv8-large-seg | 57.37(±1.38) | 77.63(±3.36) | 67.77(±2.55) | 58.03(±1.37) | 57.93(±1.76) | 78.97(±2.97) | 69.13(±2.84) | 56.57(±1.59) |
| | OSP-YOLOv8-large-seg | **58.13**(±1.02) | **79.37**(±1.02) | **69.07**(±0.84) | **58.07**(±0.06) | **59.03**(±1.17) | **80.47**(±0.80) | **70.63**(±0.96) | **56.73**(±0.35) |
| | YOLO11-large-seg | 60.37(±1.07) | 75.30(±0.95) | 68.00(±1.70) | 58.00(±0.95) | 61.30(±1.21) | 77.00(±2.33) | 69.33(±1.66) | 56.73(±0.76) |
| | OSP-YOLO11 L | **61.18**(±1.26) | **82.05**(±2.62) | **68.95**(±0.49) | **59.64**(±0.92) | **62.59**(±0.35) | **82.65**(±3.46) | **70.05**(±0.35) | **56.80**(±0.85) |
| MSCOCO | YOLOv8-large-seg | 71.43(±0.86) | 61.74(±1.04) | 67.49(±0.03) | 51.28(±0.05) | 71.30(±0.13) | 60.21(±0.49) | 64.93(±0.07) | 42.05(±0.00) |
| | OSP-YOLOv8-large-seg | **71.93**(±1.24) | **62.45**(±0.76) | **67.72**(±0.07) | **51.43**(±0.08) | **71.71**(±1.52) | **60.60**(±0.76) | **65.15**(±0.13) | **42.14**(±0.03) |
| | YOLO11-large-seg | 71.95(±0.35) | 62.55(±0.35) | 67.90(±0) | 51.50(±0.14) | 71.30(±0.42) | **60.85**(±0.35) | 65.30(±0.14) | **42.35**(±0.07) |
| | OSP-YOLO11-large-seg | **72.30**(±0.21) | **62.60**(±0.29) | **68.0**(±0.26) | **51.50**(±0.09) | **72.10**(±0.34) | 60.70(±0.20) | **65.40**(±0.15) | 42.30(±0.10) |

61.30 to 62.59, recall from 77.00 to 82.65, mAP50 from 69.33 to 70.05, and mAP50-95 from 56.73 to 56.80. On MSCOCO, the gains are more modest: applying OSP layer improves box precision, recall, and mAP50 from 71.95/62.55/67.90 to 72.30/62.60/68.00, while box mAP50-95 remains unchanged at 51.50. For mask prediction, OSP improves precision and mAP50 from 71.30 and 65.30 to 72.10 and 65.40, but shows slight decreases in recall and mAP50-95 from 60.85 and 42.35 to 60.70 and 42.30. Overall, these results suggest that OSP provides clear gains on CarParts and competitive, mostly positive improvements on MSCOCO for YOLO11-large.

## B.2. OSP layer Impact on Two-Stage Detection Model

**Experimental Settings.** We implement a two-stage detection pipeline based on the Fast R-CNN architecture (Girshick, 2015) designed for multi-task learning, specifically targeting object classification, horizontal bounding box regression, and instance segmentation. The backbone architectures (ResNet 18 and OSP-ResNet 18) are configured to extract hierarchical feature maps from four residual stages. These multi-scale features are subsequently aggregated by a Feature Pyramid Network (FPN), which generates a pyramid with strides of $[4, 8, 16, 32, 64]$. Instead of utilizing a separate Region Proposal Network (RPN), we use the annotated GT boxes as initial proposals. During training, we apply spatial jittering to learn robust refinement. The RoIAlign layer then extracts precise features from the FPN levels based on these perturbed proposals to feed the box and mask branches.

The models are trained for 100 epochs with a batch size of 32, including an initial linear warm-up phase for the first 200 iterations. We employ AdamW optimizer with a momentum of 0.9, a learning rate of $4 \times 10^{-3}$, a weight decay of 0.01, and a gradient clipping norm of 1.0. We conduct all experiments using four 3090Ti GPUs under a Distributed Data Parallel environment, utilizing 16-bit mixed precision.

**Results.** Table 11 shows that OSP yields larger growth rates in the two-stage ResNet18-FPN model than in the one-stage YOLOv8-large-seg model. On CarParts, OSP-YOLOv8-large-seg improves YOLOv8-large-seg by +1.92% and +0.07% for Box mAP50 and mAP50-95, and by +2.17% and +0.28% for Mask mAP50 and mAP50-95, respectively. In contrast, OSP-ResNet18-FPN improves ResNet18-FPN by +3.59% and +2.49% for Box mAP50 and mAP50-95, and by +4.00% and +1.85% for Mask mAP50 and mAP50-95, respectively. This suggests that OSP is more effective when combined with the stronger structural bias of a two-stage detector.

*Table 11.* Two-stage detection model on CarParts. Values denote mean(±std.)$_{\text{growth rate}}$.

| Dataset | Model | Box | | Mask | |
|---|---|---|---|---|---|
| | | mAP50 | mAP50-95 | mAP50 | mAP50-95 |
| CarParts | YOLOv8-large-seg | 67.77(±2.55) | 58.03(±1.37) | 69.13(±2.84) | 56.57(±1.59) |
| | OSP-YOLOv8-large-seg | **69.07**(±0.84)$_{+1.92\%}$ | **58.07**(±0.06)$_{+0.00\%}$ | **70.63**(±0.96)$_{+2.17\%}$ | **56.73**(±0.35)$_{+0.28\%}$ |
| CarParts | ResNet18-FPN | 63.07(±2.01) | 62.86(±1.91) | 62.24(±1.74) | 44.41(±1.18) |
| | OSP-ResNet18-FPN | **65.34**(±0.98)$_{+3.59\%}$ | **64.43**(±0.19)$_{+2.49\%}$ | **64.73**(±0.57)$_{+4.00\%}$ | **45.23**(±0.19)$_{+1.85\%}$ |

## B.3. Extended Visual Comparisons of Detection and Segmentation

To visually evaluate the effectiveness of the OSP layer, we provide a qualitative comparison between the baseline YOLOv8-large-seg and the proposed OSP-YOLOv8-large-seg.

**Classification Stability and Error Correction**    The first three rows of figure 8 illustrate cases where the OSP layer effectively corrects classification instabilities inherent in the baseline model. In the first row, the baseline model misidentifies a fork as a knife, whereas the proposed model correctly identifies the object with high confidence. A similar semantic error is observed in the second row, where YOLOv8 L misclassifies an oven as a refrigerator. The OSP-integrated model successfully resolves this ambiguity, providing the correct oven label. Furthermore, as shown in the third row, the proposed model corrects a classification error in a challenging environment where the baseline identifies a car as an airplane.

**Precision in Segmentation and Delineation**    The OSP layer significantly enhances the model's ability to delineate complex object boundaries. In the fourth row, OSP-YOLOv8-large-seg achieves superior segmentation precision for intricate objects such as clock and knife by more accurately capturing their spatial boundaries. As shown in the mural scene in the fifth row, the baseline merges overlapping train and smoke regions into a single mask, whereas the proposed model successfully separates them into distinct semantic instances. This demonstrates that the OSP layer effectively refines feature representation, ensuring robust performance in complex scenarios.

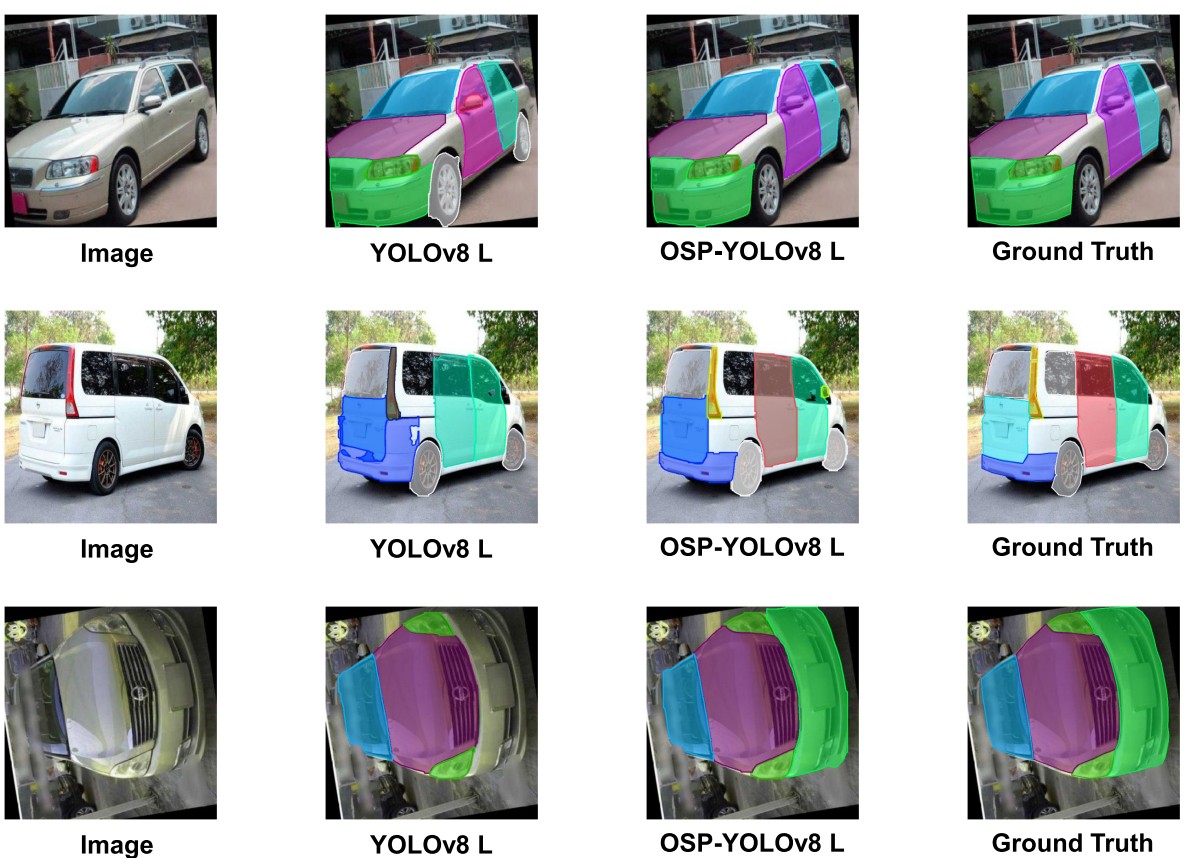

*Figure 7.* Qualitative comparison on the CarParts dataset. The proposed OSP-YOLOv8 L demonstrates more precise boundary definition and label consistency for individual automotive components compared to the baseline.

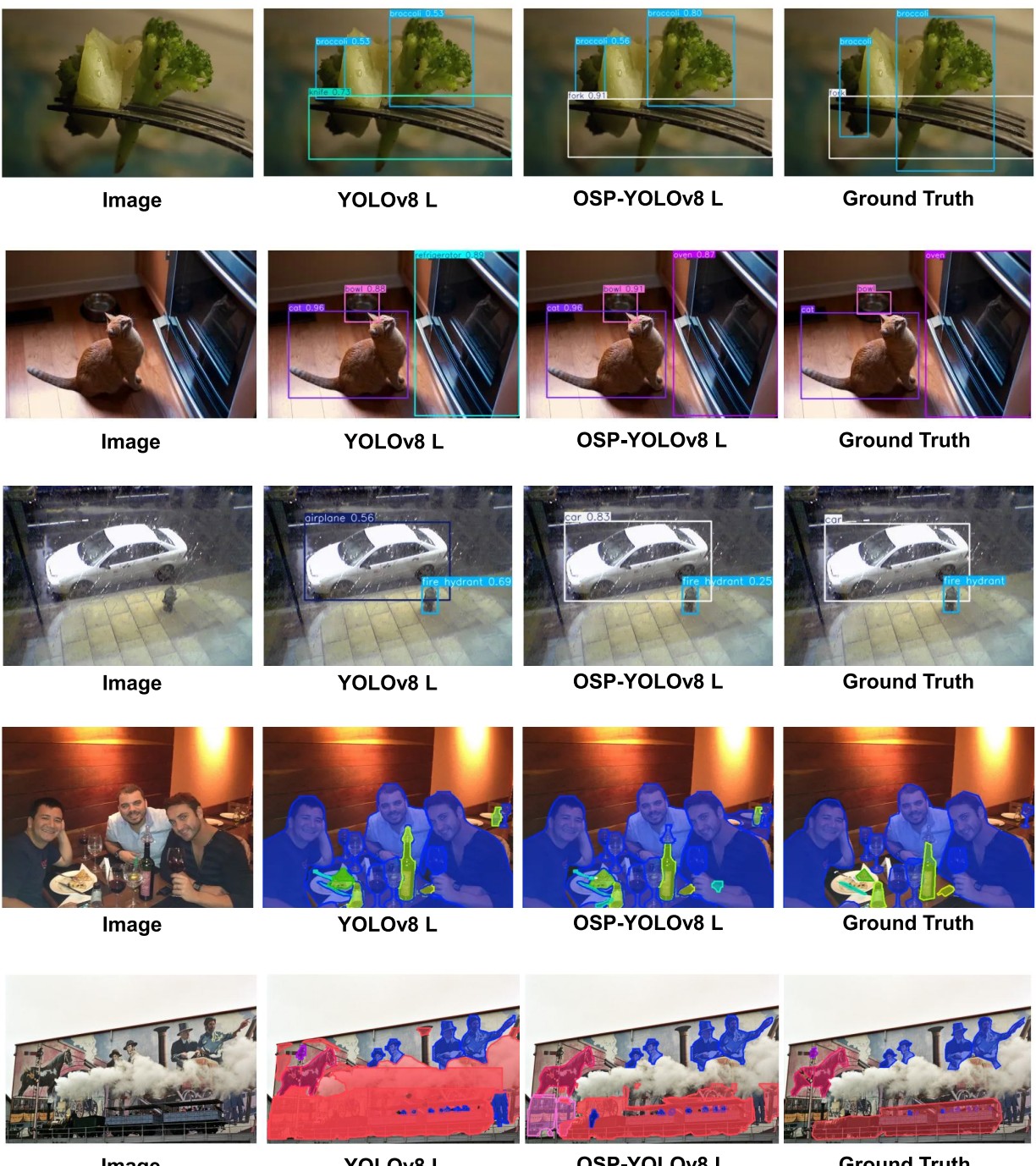

*Figure 8.* Qualitative results of the baseline and proposed models on MS COCO.

## C. Generalization on Out-of-Distribution (OOD)

(Zhu et al., 2026) focuses on orthogonality across domains to disentangle different domains for OOD generalization. Differently, we disentangle independent factors for generalization rather than domains for distribution shift. However, Disentanglement learning is utilized for OOD generalization, we then validate the OSP in the OOD environment. To validate OOD generalization, we conduct OOD experiments on OOD-classification (Gulrajani & Lopez-Paz, 2021), and OOD-detection (Mao et al., 2023).

**OOD-Classification.** Table 12 reports OOD classification results on four DomainBed benchmarks. Overall, OSP-ResNet18 achieves a slightly better average accuracy than ResNet18 (64.43 vs. 64.30), indicating that OSP provides a modest but consistent benefit under distribution shift. In particular, OSP improves performance on VLCS (75.24 vs. 75.18), TerraIncognita (44.50 vs. 43.19), and OfficeHome (57.67 vs. 56.54), while showing a small drop on PACS (80.30 vs. 82.31). These results suggest that the structural bias introduced by OSP can enhance robustness and cross-domain generalization, although the gains are modest and dataset-dependent.

*Table 12.* OOD-classification performance on various DomainBed dataset.

|  | PACS | VLCS | Terraincognita | Officehome | Avg. |
|---|---|---|---|---|---|
| ResNet18 | **82.31**($\pm$7.96) | **75.18**($\pm$13.81) | 43.19($\pm$6.80) | 56.54($\pm$9.59) | 64.30($\pm$17.78) |
| OSP-ResNet18 | 80.30($\pm$10.16) | **75.24**($\pm$14.77) | **44.50**($\pm$8.10) | **57.67**($\pm$11.17) | **64.43**($\pm$16.45) |

**OOD-Detection.** Table 13a and 13b report the OOD detection results on DetectBench and COCO-O, respectively. Overall, OSP-YOLOv8-large consistently improves the average OOD performance over YOLOv8-large on both benchmarks, increasing the average score from 32.66 to 33.25 on DetectBench and from 27.50 to 27.79 on COCO-O. On DetectBench, OSP improves the results in almost all corruption types, including noise, blur, weather, and digital, while remaining nearly identical on 3D. On COCO-O, OSP achieves better performance on cartoon, handmade, sketch, tattoo, and weather, with only a slight drop on painting. These results suggest that OSP provides a consistent robustness benefit under distribution shift and improves OOD generalization in object detection across diverse corruption and style-shift scenarios.

*Table 13.* OOD-detection performance on various benchmarks.

*(a)* DetecBench.

|  | iid | noise | blur | weather | digital | 3D | Avg. |
|---|---|---|---|---|---|---|---|
| YOLO8v8-large | 51.25 | 30.31($\pm$1.33) | 20.44($\pm$9.00) | 38.49($\pm$7.36) | 32.99($\pm$12.65) | 38.72($\pm$7.85) | 32.66($\pm$11.28) |
| OSP-YOLOv8-large | **51.37** | **30.43**($\pm$0.92) | **20.67**($\pm$9.07) | **38.73**($\pm$7.27) | **34.82**($\pm$11.33) | **38.71**($\pm$7.90) | **33.25**($\pm$10.98) |

*(b)* COCO-O.

|  | cartoon | handmake | painting | sketch | tatoo | weather | Avg. |
|---|---|---|---|---|---|---|---|
| YOLO8v8-large | 22.92($\pm$0.69) | 27.54($\pm$0.25) | **40.32**($\pm$0.41) | 20.69($\pm$1.22) | 14.62($\pm$1.15) | 38.90($\pm$0.28) | 27.50($\pm$14.15) |
| OSP-YOLOv8-large | **23.39**($\pm$0.17) | **27.98**($\pm$0.56) | 40.10($\pm$0.26) | **20.94**($\pm$0.23) | **15.14**($\pm$0.91) | **39.16**($\pm$0.76) | **27.79**($\pm$14.32) |

