# OpenReview forum: "Layer-Centric Factors of Variation Disentanglement for Task- and Model-Agnostic Generalization"
_ICML.cc/2026/Conference — ICML 2026 regular_

### Official Review · Reviewer_qU9s · 2026-03-11

**Soundness:** 2
**Presentation:** 1
**Significance:** 2
**Originality:** 2
**Overall Recommendation:** 4
**Confidence:** 4

**Summary:**

This paper proposes the Orthogonal Subspaces Projection (OSP) layer for disentangled representation learning, which regularizes the output of the intermediate layers to be mutually orthogonal. Unlike prior work which often focuses on applying regularization on the last layer of the network, OSP layer additionally provides a lightweight way to induce an inductive bias for disentanglement on intermediate layers. To validate the effectiveness of the proposed method, the authors applied the OSP layers on top of existing methods and the proposed method showed consistent improvement on various tasks including disentanglement task, several computer vision tasks and NLP tasks.

**Compliance With Llm Reviewing Policy:**

Affirmed.

**Final Justification:**

Although the proposed method is not explicitly designed to achieve disentanglement on its own and remains insufficient as a standalone disentanglement mechanism, it provides a useful additional inductive bias that empirically provides consistent improvements across different models and settings. Thus, I decided to raise my score to a weak accept.

**Key Questions For Authors:**

- What is the definition of disentanglement used in this paper, and why should orthogonal subspaces promote it?

- What is the main claim of the paper? Please refer to the second item in the strengths&weaknesses section.

- Is the proposed OSP layer intended to work as a standalone module for disentanglement, or required to use on top of existing disentanglement objectives?

- In Figure 4, Could the authors provide qualitative comparison to ResNet without OSP layers to directly examine the effect of adding OSP layers? Without comparison, it is hard to know whether the correct Grad-CAM results are coming from ResNet itself or how OSP actually improves the Grad-CAM results.

**Limitations:**

No, I cannot find the discussions on limitation and potential negative societal impact of this work.
It is encouraged to discuss about the method’s scope and assumptions, as well as common failure cases and settings where the proposed method would be less effective.

**Strengths And Weaknesses:**

**Strengths**
- This paper explores an interesting direction in terms of architectural bias for disentanglement learning.
- The proposed method is lightweight and easy to integrate into existing models.
- The proposed method consistently improves the existing methods on various tasks such as image classification, detection, segmentation, and NLP tasks. It is interesting to see simple modification leads to consistent improvement over various tasks.

**Weaknesses**
- **Unclear definition of disentanglement and its connection to orthogonal subspace**:
A major concern is that the paper does not provide a clear definition of disentanglement, and therefore it is difficult to assess whether the orthogonal subspace promotes “disentanglement”.  In addition, the following statement needs more justification: “Motivated by disentanglement learning methods that conduct objectives on each dimension of latent vectors to be independent, we enforce divided subspaces to be orthogonal to encourage independent effect of disentanglement learning methods.”. This is not a trivial statement and the authors should clarify this. Prior work often defines disentanglement with statistical independence and promotes it through learning objectives.
In contrast, this paper imposes orthogonal subspaces but does not clearly state how they define the disentanglement and how orthogonal subspace would achieve such definition.
For example, how does the orthogonal subspace induce statistical independence or how does the orthogonal subspace guarantee that a change of individual latent lead to a change of corresponding factor in output image space leaving other factors unchanged?
Since this is related to the main insight and motivation of this work, a more detailed clarification would strengthen this work.

- **Main claim of this paper is unclear**: The main claim of this paper is somewhat confusing. Is it (1) to resolve dependency of inductive bias upon learning objectives and thereby propose a new architectural bias or (2) to propose that regularization for disentanglement should be applied on intermediate layers rather than just the final last layer?

- **The terminology is ambiguous and difficult to understand**: The meaning of the terms “latent-space objective” (L55, L156, L125 right paragraph), “ latent-vector-centric objective” and “layer-centric” (throughout the paper) are ambiguous and quite confusing. These are not commonly used terms and it is still confusing even after reading the whole manuscript. Do latent-vector-centric/layer-centric mean whether to apply regularization on the last or intermediate layer of the representation? If so, it is recommended to choose more direct and clearer terminology.

- **The role of the proposed OSP layer is not clear**: In the experiment section, the OSP layers are trained on top of existing disentanglement learning work such as beta-vae and beta-TCVAE. Does the proposed OSP layer work as a standalone module, or is it intended to enhance disentanglement when combined with prior disentanglement learning frameworks? If the latter is the case, the current form of overall writing is somewhat misleading. For example, in the first paragraph of the experiment section, the authors state that “we train models on disentanglement learning benchmarks to **quantitatively assess whether the equipped OSP layer model separates Factors of Variation (FoV)** and …“.
It gives the impression that the OSP layer itself is sufficient to promote disentanglement.
If the proposed OSP layer alone is sufficient to promote disentanglement, quantitative results should be provided and compared to the baselines.

- **The empirical validation for disentanglement is limited to VAE-based methods**: In experimental results regarding the comparison to existing disentanglement methods, the authors only evaluated VAE-based methods. If the proposed method is truly a plug-and-play module, it should be examined across more various architectures such as GAN-based method [1] or Diffusion-based methods [2, 3].

**References**

[1] Lin et al., InfoGAN-CR and Model Centrality: Self-supervised Model Training and Selection for Disentangling GANs, in ICML 20.

[2] Yang et al, DisDiff: Unsupervised Disentanglement of Diffusion Probabilistic Models, in NeurIPS 23.

[3] Yang et al., Diffusion Model with Cross Attention as an Inductive Bias for Disentanglement, in NeurIPS 24.

---

> ### Author Rebuttal · Authors · 2026-03-31
>
> Thanks to Reviewer qU9s for the fruitful comments. We use **W, Q**, and **L** to denote weaknesses, questions, and limitations, respectively. **Due to the rebuttal length limitation, we kindly ask to download and check the revised paper: https://anonymous.4open.science/r/OSP_revised_paper-7F46/**.
>
> - Ans. of W1-1, W1-2, and Q1
> >**Definition of Disentanglement and Connection to Orthogonality**: 1) We follow the dimension-wise disentangled representation [1], where different latent dimensions are encouraged to capture distinct underlying factors of variation [1]. 2) From this perspective, prior disentanglement methods often encourage reduced statistical dependence across latent dimensions, for example, by penalizing total correlation [2,3], and propose objectives that encourage orthogonality across dimensions [4] with a loss function. Motivated by previous works, we explicitly enforce orthogonality among subspaces from a structural perspective rather than proposing a new loss function.
> >
> > [1] Disentangled Representation Learning, TPAMI 2024.
> >
> > [2] Isolating Sources of Disentanglement in VAEs, NeurIPS 2018.
> >
> > [3] Disentangling by factorising, ICML 2018.
> >
> > [4] CFASL: Composite Factor-Aligned Symmetry Learning for Disentanglement in Variational AutoEncoder, TMLR 2024.
>
> - Ans. of W 1-3, W5, and Q3.
> >**(W1-3, Q5)** First, our claim is not that an orthogonal subspace alone, without any objectives, can achieve disentanglement learning. Rather, we suggest that the OSP layer improves disentanglement learning when combined with appropriate objectives, and we demonstrate this using VAE-based models, since these already encourage latent dimensions to be statistically independent through the KL divergence term.
> >
> >In addition, **we discuss the relation** between orthogonal subspace and disentanglement learning at **Reviewer eaHB Ans. of Q1-2 and Q1-3**.
> >
> >**(Q3. Pure Architectural Advantage)**: **We add more results on Table 6 in the revised paper (page 15)**. Although the OSP does not by itself promote disentanglement, we show that applying the OSP layer to AE (OSP-AE) still provides structural benefits for disentanglement learning compared to AE. To address pure architectural benefit, we provide empirical evidence under no disentanglement regularization, then we compare AE and OSP-AE (i.e., without disentanglement regularization).
> \\begin{array}{|c|c|c|}
> \\hline
> 3D Shapes& AE&OSP-AE\\\\
> \\hline
> beta-VAE&67.20(\pm2.28) &\textbf{76.80}(\pm5.02)\\\\
> \\hline
> \\end{array}
> As shown above, OSP-AE outperforms AE. This suggests that the proposed OSP structure itself provides a useful inductive bias for learning more disentangled representations.
> **We revise the manuscript to avoid this misunderstanding.**
>
> - Ans. of W2 and Q2.
> > Our claim is not that OSP removes the need for a disentanglement loss function. Rather, OSP complements these objectives by introducing a structural bias in intermediate layers, thereby distributing the disentanglement burden beyond the final bottleneck layer. This helps reduce cross-factor interference and improves disentanglement (Figure 3 and Table 1 in page 5).
>
> - Ans. of W3.
> > We agree with the reviewer’s point that may confuse readers. To alleviate it, we will change the terms to indicate a more direct meaning in the whole manuscript, including the paper title. In the original manuscript, *latent-space objective* and *latent-vector-centric objective* referred to **prior methods** that impose disentanglement losses **on latent representations**, whereas *layer-centric* referred to a structural approach, such as OSP, rather than an additional loss. To avoid confusion for other reviewers in the revised version, if given the opportunity.
>
> - Ans. of W4.
> > As clarified in our responses to **W2, W3, and Q2**, our method is intended to enhance disentanglement when combined with existing frameworks. We revised the manuscript accordingly to avoid suggesting that OSP alone promotes disentanglement.
>
> - Ans. of W6.
> > **To address the limitation of VAE-only validation**, we further evaluate OSP on DisDiff and IB-GAN [8]. We choose IB-GAN over InfoGAN-CR because it is more recent, and report the original metrics of each paper for fairness. **Also, we add more results to Table 7 in the revised paper (page 16)**.
> As shown below, applying OSP consistently improves performance over the corresponding baselines.
> \\begin{array}{|c|c|}
> \\hline
> 3D Shapes&Factor VAE score\\\\
> \\hline
> DisDiff&90.20(\pm4.30)\\\\
> OSP-DisDiff&\textbf{92.00}(\pm3.34)\\\\
> \\hline
> \\end{array}
> \\begin{array}{|c|c|}
> \\hline
> CdSprites&Factor VAE score\\\\
> \\hline
> IB-GAN&75.10(\pm5.16)\\\\
> OSP-IB-GAN&\textbf{78.49}(\pm3.65)\\\\
> \\hline
> \\end{array}
> >
> > [8] IB-GAN: Disentangled Representation Learning with Information Bottleneck Generative Adversarial Networks, AAAI 2021.
>
> - Ans. of Q4.
> > We revise Figure 4 to provide a direct qualitative comparison between ResNet and OSP-ResNet on the same input images. Please see the revised manuscript.

---

> > ### Author Rebuttal · Reviewer_qU9s · 2026-04-04
> >
> > I thank the authors for additional experiments and detailed clarifications.
> > In particular, consistent improvement in non-VAE frameworks (DisDiff and IB-GAN) suggests that its effectiveness is not limited to VAE frameworks.
> > It is also now clear that the OSP layer should not be viewed as a standalone bias for disentanglement, but rather as an additional inductive bias applied at intermediate layers, which is orthogonal to existing methods that impose disentanglement objectives at the final output layer.
> > Although the formal definition of disentanglement is still not clearly stated, it is empirically demonstrated that applying orthogonal subspace constraints in intermediate layers improve the the disentanglement metrics and downstream task performances.
> > That said, it could be a limitation that the method is not explicitly designed to achieve disentanglement by itself, and OSP-AE results indicate that the OSP layer alone is insufficient as a standalone disentanglement method (DCI-D score is very low.) Nevertheless, when viewed as an additional inductive bias that provides consistent improvements across different settings, the overall contribution seems meaningful.
> > Thus, I decided to raise my score to a weak accept.

---

### Official Review · Reviewer_3vnQ · 2026-03-12

**Soundness:** 3
**Presentation:** 3
**Significance:** 3
**Originality:** 3
**Overall Recommendation:** 4
**Confidence:** 2

**Summary:**

This paper proposes the orthogonal subspace projection (OSP) layer, which projects latent features into mutually orthogonal subspaces. The proposed OSP is evaluated across several tasks including both CV and NLP. Experiments show its benefits but relatively modest improvements across multiple models and tasks.

**Compliance With Llm Reviewing Policy:**

Affirmed.

**Key Questions For Authors:**

1. Whether the orthogonal feature is general for both CV and NLP tasks?
2. It would be great if authors can provide more experimental evidence for my concerns presented in weaknesses.

**Limitations:**

I didn't find any discussion about the limitation.

**Strengths And Weaknesses:**

Strengths
+ The proposed method is model-agnostic and plug-and-play, making it applicable to CV and NLP models.
+ The paper includes experiments across multiple domains and tasks, including disentanglement benchmarks, computer vision tasks, NLP tasks, and parameter-efficient fine-tuning.

Weaknesses
- The method assumes that different factors of variation can be separated into mutually orthogonal subspaces, which may not always hold in practice. I guess it depends on data distributions. Some dataset may well align with the orthogonal subspace decomposition, but others may not. If authors claim it's a general feature, especially for NLP, some theoretical or empirical evidence is needed. Otherwise, this limitation should be discussed.
- The paper lacks a sufficiently detailed ablation study on important hyper parameters, such as: number of subspaces, subspace dimension, and which layers benefit most. Without such analysis, it is difficult to understand which design choices are critical to the observed performance gains.
- Experiments on NLP tasks seem weak, with only a small model BERT and single dataset. Results on popular NLP benchmarks are desired. Although the results presented in Sec. 4.4 have some improvement, I highly doubt whether such benefits can be still obtained when applying OSP to larger models and more datasets.

---

> ### Author Rebuttal · Authors · 2026-03-31
>
> Thanks to Reviewer 3vnQ for the fruitful comments. We use **W, Q**, and **L** to denote weaknesses, questions, and limitations, respectively. **Due to the rebuttal length limitation, we kindly ask to download and check the revised paper: https://anonymous.4open.science/r/OSP_revised_paper-7F46/**.
>
> - Ans. of W1.
> > We agree with the reviewer’s perspective that orthogonality is dependent on the dataset. However, in recent work **[1]** explicitly argues that **partial orthogonality in the embedding space naturally encodes semantic independence**, provides an algebraic encoding of that structure, and supports practical tools for analyzing embeddings, as demonstrated on CLIP text embeddings. Similarly, **[2]** argues in **Theorem 8 that hierarchical semantic structure is encoded through orthogonality between representations in LLMs**. In this sense, our claim is not that all factors are universally orthogonal, but **rather that structurally encouraging such geometry can be beneficial** when the underlying factors or semantic relations are compatible with it. We further support this point empirically by adding GLUE results in **Ans. of W3, where OSP also improves downstream NLP performance (GLUE)**.
>
> - Ans. of W2.
> > We response to ablation study on hyper parameters on **reviewer 93MA Ans. of Q1**.
>
> - Ans. of W3.
> >**(Experiment on GLUE task)**: To address the lack of NLP experiments, we add the GLUE task with large models (BERT, BERT-large, and RoBERTa-large) and **we add results on Table 10 in the revised paper, including all 9 tasks (Table 10 in page 20)**.
> \\begin{array}{|c|c|c|c|c|c|c|c|}
> \\hline
> \text{GLUE} & \text{Param.} & \text{STS-B} & \text{MRPC} & \text{SST-2} & \text{QQP} &\text{QNLI} & \text{MNLI}\\\\
> \\hline
> \text{BERT-base} & 109.48M & 86.29/86.08 & 81.37/87.21 & 92.32 & 90.51/87.31 & 90.66 & 83.91/84.10\\\\
> \text{OSP-BERT-base} & 109.63M & \textbf{86.38}/\textbf{86.40} & \textbf{81.62}/\textbf{87.44} & \textbf{92.43} & \textbf{84.52}/\textbf{84.48} & \textbf{91.16} & \textbf{84.52}/\textbf{84.48} \\\\
> \\hline
> \text{BERT-large} & 335.14M & 87.38/87.49 & 81.62/87.09 & 92.78 & 91.31/88.40 &\textbf{92.26} & 86.34/86.44 \\\\
> \text{OSP-BERT-large} & 335.40M & \textbf{88.36}/\textbf{88.38} & \textbf{82.60}/\textbf{87.80} & \textbf{93.00} & \textbf{91.42}/\textbf{88.50} & 92.22 &\textbf{86.68}/\textbf{86.75}\\\\
> \\hline
> \text{RoBERTa-large} & 355.43M & 91.09/91.17 & 87.99/91.51 & 95.99 & 90.74/90.48 &94.33 &90.74/90.48 \\\\
> \text{OSP-RoBERTa-large} & 355.76M & \textbf{91.25}/\textbf{91.34} & \textbf{88.73}/\textbf{91.90} & \textbf{96.33} & \textbf{91.91}/\textbf{89.30} & \textbf{94.42} &\textbf{90.98}/\textbf{90.76} \\\\
> \\hline
> \\end{array}
>
> - Ans. of Q1.
> > We do not claim that orthogonality is a universal property across either CV or NLP tasks. Rather, our claim is that orthogonal structure can serve as a useful inductive bias when the underlying factors are approximately separable.
> >
> > **In CV**, we discuss this perspective **in Sections 2.1 and 2.2 of the manuscript**, mainly in the context of disentanglement learning and classification. Prior work in disentanglement often assumes that different factors can be separated in the representation space, and orthogonality-based projections have also been used in classification settings. **These observations suggest that orthogonal structure can be useful in vision, although its effectiveness may depend on the data distribution**.
> >
> > For NLP, we do not claim that orthogonality is a settled general property. Rather, **recent studies [1,2] suggest that orthogonality-related structure can emerge in large embedding spaces and language models**, indicating that this is a promising direction. We believe this is particularly plausible in NLP, where **semantic and hierarchical structure are often reflected** in learned representations. In this sense, orthogonal structure may serve as a useful inductive bias when such relationships are approximately captured in the feature space, although its effectiveness may still depend on the task and data distribution. We will clarify this scope and limitation in the revised manuscript.
>
>
> ---
> >[1] Uncovering Meanings of Embeddings via Partial Orthogonality, NeurIPS, 2023.
> >
> >[2] THE GEOMETRY OF CATEGORICAL AND HIERARCHICAL CONCEPTS IN LARGE LANGUAGE MODELS, ICLR, 2025.

---

> > ### Author Rebuttal · Reviewer_3vnQ · 2026-04-03
> >
> > The rebuttal seems reasonable, and I'm still positive to this paper.

---

### Official Review · Reviewer_eaHB · 2026-03-12

**Soundness:** 2
**Presentation:** 3
**Significance:** 2
**Originality:** 3
**Overall Recommendation:** 4
**Confidence:** 4

**Summary:**

This paper proposes the Orthogonal Subspaces Projection (OSP) layer for disentanglement learning. Unlike latent-vector-centric methods that constrain bottleneck features, OSP integrates into intermediate layers and promotes factor-of-variation separation by projecting features into orthogonal subspaces. Experiments across vision and NLP tasks show that OSP improves both disentanglement quality and downstream generalization.

**Compliance With Llm Reviewing Policy:**

Affirmed.

**Final Justification:**

My concerns have been adequately addressed, and I will raise my score to 4.

**Key Questions For Authors:**

1. Could the authors provide simulation results or theoretical evidence demonstrating that the proposed Layer-Centric Disentanglement Learning consistently outperforms latent-vector-centric disentanglement methods.

2. The discussion of recent related works on disentanglement learning is limited. Relevant references that should be discussed include:

   [1] Architecture Disentanglement for Deep Neural Networks, ICCV 2022

   [2] Disentangled Representation Learning, TPAMI 2024

   [3] CFSM: A Novel Causal Feature Selection Module for Two-Dimensional Out-of-Distribution Generalization, TPAMI 2026

3. Since disentanglement is often motivated by improvements in model robustness and generalization, it is recommended to include experiments on OOD generalization benchmarks, such as DomainBed[4] (image classification) and DetectBench[5] (object detection).

   [4] In Search of Lost Domain Generalization, ICLR 2021

   [5] DetectBench: An Object Detection Benchmark for OOD Generalization Algorithms

4. The experimental improvements reported in Table 3 are relatively limited compared to results on other tasks. The authors should provide an explanation or analysis to clarify why the performance gains are modest in this setting and what insights can be drawn from it.

**Limitations:**

The paper proposes an interesting Layer-Centric Disentanglement Learning approach, but its advantages over latent-vector-centric methods are not clearly justified, lacking theoretical or simulation evidence. Related work on disentanglement is under-discussed, and experiments do not include key OOD generalization benchmarks such as DomainBed or DetectBench.

If the author could address my concerns, I am willing to revise my score.

**Strengths And Weaknesses:**

Strengths:
The paper is well-written and easy to follow. The proposed method is interesting and appears to be novel. The proposed method is validated across various tasks.


Weaknesses: The paper proposes an interesting Layer-Centric Disentanglement Learning approach, but its advantages over latent-vector-centric methods are not clearly justified, lacking theoretical or simulation evidence. Related work on disentanglement is under-discussed, and experiments do not include key OOD generalization benchmarks such as DomainBed or DetectBenc

---

> ### Author Rebuttal · Authors · 2026-03-31
>
> Thanks to Reviewer eaHB for the fruitful comments. We use **W, Q**, and **L** to denote weaknesses, questions, and limitations, respectively. **Due to the rebuttal length limitation, we kindly ask to download and check the revised paper: https://anonymous.4open.science/r/OSP_revised_paper-7F46/**.
>
> - Ans. of Q1 and W.
> >**1. Controlled simulation results (Table 1 on revised manuscript)**: We revise the presentation of Table 1 from the submitted version for clarity while preserving the original results and denote models with OSP layers as OSP-{model name}. Table 1 already shows **two key findings**. 1) First, under the same loss function, applying OSP layers to standard layers consistently **improves disentanglement performance in most cases** on 3D Shapes and MPI3D dataset. 2) Second, Table 1 also shows that adding a layer-centric bias can be **more effective than relying only on stronger latent-vector-centric objectives**, since OSP-$\beta$-VAE is competitive with or better than standard $\beta$-TCVAE and CLG-VAE on several metrics.
> >
> >**2. Structural explanation: reduced cross-subspace interference (Section 3.1)**: The OSP layer provides a structural mechanism that can **reduce cross-factor interference more directly than a loss function imposed** only at the latent vector space. As described in Section 3.1, OSP projects intermediate activations $v$ into $K$ mutually orthogonal subspaces. The output for the $j$-th subspace is defined as $z_j=Q_j^\top v$. Under the natural assumption that a factor-specific perturbation $\Delta v_i$ aligns with its corresponding subspace $\mathcal{S}_i$, the response in any other subspace $j$ ($j\neq i$) becomes $\Delta z_j = Q_j^\top \Delta v_i=0$.
> Because the constructed subspaces are orthogonal ($\mathcal{S}_i \perp \mathcal{S}_j$), this provides a principled explanation for why OSP can reduce cross-subspace leakage. While this is not intended as a universal theorem covering every possible setting, **it offers a structural explanation for why OSP can mitigate the entanglement leakage that may persist in standard layers**.
> >
> >**3. Geometric evidence consistent with this explanation (Figure 3)**: Figure 3 further supports the above *structural explanation*. Compared with standard $\beta$-TCVAE, OSP-$\beta$-TCVAE shows lower off-diagonal cosine similarity (Eq. 5-7 in manuscript) between factor-direction vectors, indicating weaker coupling across factors. This geometric result is consistent with the reduced cross-subspace interference induced by the proposed structure.
>
> - Ans. of Q2, W and L.
> > We discuss [1] in Line 49, [2] in Line 55, and [3] in Line 1074. Due to the text limitation, we kindly ask you to check the revised paper.
>
> - Ans. of Q3, W and L.
> > We add additional OOD generalization experiments on DomainBed and on COCO-O [6] for object detection to **Tables 11 and 12 in the revised paper (Line 1087 and 1103)**. For object detection, we used COCO-O [6] as a **published benchmark** for natural distribution shifts that is directly compatible with our current detection setup. As shown in the tables below, applying OSP layer, OSP-{ResNet18 (R18), YOLO11 (Y11)}, outperforms baselines. These gains are modest, but they consistently support the robustness benefit of OSP under distribution shift.
> \\begin{array}{|c|c|}
> \\hline
> &Terra&Office.&Avg.(whole ~dataset)\\\\
> \\hline
> R18 & 43.19&56.54&64.30\\\\
> OSP-R18&\textbf{44.50}&\textbf{57.67}&\textbf{64.43}\\\\
> \\hline
> \\end{array}
> \\begin{array}{|c|c|}
> \\hline
> &[6]\\\\
> \\hline
> Y11&27.50(\pm14.15)\\\\
> OSP-Y11&\textbf{27.79}(\pm14.32)\\\\
> \\hline
> \\end{array}
> >
> >[6] COCO-O: A Benchmark for Object Detectors under Natural Distribution Shift. CVPR 2023.
>
> - Ans. of Q4.
> > **(Explanation of modest improvement in a one-stage architecture)**: We first add detection **results of YOLO11 on Table 8 in the revised paper (page 16)**. To explain the relatively modest gain of OSP-YOLO (OSP-Y) over YOLO (Y) in Table 3, we apply OSP on a two-stage detector, ResNet18+FPN (R18F), and **more results are in Table 9 in the revised paper (page 17)**. The mAPs growth rate between OSP-Y8 and Y8 on CarParts (CP) dataset from Table 3 is **1.92** from (67.77 to 69.07) and **0.00** (from 58.03 to 58.07). And as shown below, the improvement brought by OSP is consistently larger in the two-stage setting than in YOLO-v8 **($3.59>1.92$, and $2.49>0.00$)**.  This suggests that the more modest gain in Y8 may stem from the one-stage architecture, where the structural bias is less explicit. By contrast, the larger gains in R18F indicate that **OSP is more effective when combined with the stronger structural bias of a two-stage detector**.
> \\begin{array}{|c|c|c|}
> \\hline
> CP&box/mAP50&box/mAP50-95\\\\
> \\hline
> R18F&63.07(\pm2.01)&62.86(\pm1.91)\\\\
> OSP-R18F&65.34(\pm0.98)&64.43(\pm0.19)\\\\
> \hline
> growth~rate&\textbf{3.59}&\textbf{2.49}\\\\
> \\hline
> \\end{array}

---

> > ### Author Rebuttal · Reviewer_eaHB · 2026-04-03
> >
> > I thank the authors for their detailed rebuttal. My concerns have been adequately addressed, and I will raise my score to 4.

---

### Official Review · Reviewer_93MA · 2026-03-12

**Soundness:** 4
**Presentation:** 4
**Significance:** 2
**Originality:** 3
**Overall Recommendation:** 5
**Confidence:** 4

**Summary:**

The work introduces the Orthogonal Subspaces Projection (OSP) layer which integrates into existing architectures to assist with factors of variation (FoV) separation. The OSP works by projecting features into mutually orthogonal subspaces. The OSP layer is introduced into existing methods, and it improves disentanglement quality consistently. The learned representation performs better on downstream tasks.

The OSP layer separates deep encodings into K mutually orthogonal subspaces and performs dimensionality compression to reduce redundancy. The OSP layer has a mxm matrix which is divided into K orthogonal weights by QR decomposition with K column groups in Q. z, the concatenated outputs of Q^T v, is used for the output representation. To avoid large amounts of computation induced by QR decomposition, they pre-compute a fixed orthogonal matrix Q (or use identity for pretrained init) and learn a shared, small block rotation matrix. The rotation matrix is parameterized via a Cayley transform to avoid expensive restraints/projections.

The block is added to B-VAE, B-TC-VAE, and CLG-VAE. It Is tested on 3D shapes, MPI3D. The method seems to consistently improve disentanglement scores by a small margin. Downstream applications seem to be consistently improved by a small margin as well.

Overall, the paper is well-written, interesting, and useful for many downstream applications. I am interested in the added computation during training and inference. Simple computation time based experiments would suffice. I am leaning towards accept.

**Compliance With Llm Reviewing Policy:**

Affirmed.

**Final Justification:**

The rebuttal addressed my main concerns. In my initial assessment, I found that this paper is well-written and that the method is useful. I believe it would be interesting to readers of ICML, so it should be accepted.

**Key Questions For Authors:**

What is the effect of number of subspaces? Is this determined by architecture, dataset, or potentially both?

How much computation does the block add?

Fig 2 seems to be missing “azimuth”.

**Limitations:**

No experiments on computation requirement of the OSP layer in real settings. It would be interesting to see a comparison of the computation for vanilla OSP (no computation optimizations with full QR) vs the efficient implementation. Some training time and inference time comparisons would do.

**Strengths And Weaknesses:**

Strengths

The method is well-motivated and explained clearly

Many experiments support the claims that the method improves downstream performance

Strong reproducibility information

Weaknesses

No experiments on computation requirement of the OSP layer in real settings. It would be interesting to see a comparison of the computation for vanilla OSP (no computation optimizations with full QR) vs the efficient implementation. Some training time and inference time comparisons would do.

---

> ### Author Rebuttal · Authors · 2026-03-31
>
> Thanks to Reviewer 93MA for the fruitful comments. We use **W, Q**, and **L** to denote weaknesses, questions, and limitations, respectively. **Due to the rebuttal length limitation, we kindly ask to download and check the revised paper: https://anonymous.4open.science/r/OSP_revised_paper-7F46/**.
>
> - Ans. of W1, Q2 and L.
> > To address computational cost, we compare our efficient implementation with the vanilla OSP using full QR decomposition (OSP + QR) with ResNet 50 under the same 512 batch size and hardware setting (RTX-3090). *Sub dim.* denotes subspace dimension size.
> \\begin{array}{|c|c|c|c|c|}
> \\hline
> \text{sub dim.} & \text{Method} & \text{Training (batch/s)} & \text{Inference (batch/s)} & \text{VRAM (GB)} \\\\
> \\hline
> & \text{Baseline} & 2.36 & 7.87 & 3.97 \\\\
> \\hline
> 32 & \text{OSP+QR} & 0.97 & 1.47 & 7.18 \\\\
> 32 & \text{Ours} & \textbf{1.69} & \textbf{3.78} & \textbf{6.01} \\\\
> \\hline
> 64 & \text{OSP+QR} & 0.87 & 1.08 & 6.92 \\\\
> 64 & \text{Ours} & \textbf{1.34} & \textbf{2.37} & \textbf{6.00} \\\\
> \\hline
> \\end{array}
> As shown above, ours consistently reduced training and inference cost across subspace dimensions. For example, at *sub dim.*=32, ours improves training throughput from 0.97 to 1.69 iter/s (1.74×) and inference throughput from 1.47 to 3.78 iter/s (2.57×), while reducing training VRAM from 7.18GB to 6.01GB.
> Similar trends hold for sub-dim=64, confirming that the proposed efficient implementation substantially mitigates the practical overhead of the full-QR version.
> >
> > Relative to the baseline ResNet-50, the efficient OSP block still introduces additional cost, but it is substantially more practical than the vanilla full-QR implementation. These results confirm that the proposed efficient implementation significantly mitigates the practical overhead of full-QR OSP in real settings. **We add this comparison to Table 14 in the revised paper (page 21)**.
>
> - Ans. of Q1.
> > To address hyper-parameter sensitivity, we conduct a sensitivity analysis with respect to both (1) the subspace dimension size (sub dim.) and (2) the number of subspaces (num. of subspace) on three GLUE tasks with different models and dataset scales: RTE (small), SST-2 (medium), and QQP (large).
> >\\begin{array}{|c|c|c|c|c||}
> \\hline
> \text{BERT (sub dim.)} & \text{RTE} & \text{SST2} & \text{QQP} (Acc.) & \text{QQP} (F1) \\\\
> \\hline
> 16 & 62.09 & 91.86 & 90.41 & 87.18 \\\\
> 32 &31.37 & 91.97 & 90.49 & 87.28 \\\\
> 64 & \textbf{65.89} & \textbf{92.43} & \textbf{90.60} & \textbf{87.43} \\\\
> 128 & 57.76 & 91.97 & 90.56 & 87.36 \\\\
> \\hline
> \text{RoBERTa (sub dim.)} & \text{RTE} & \text{SST2} & \text{QQP} (Acc.) & \text{QQP} (F1)\\\\
> \\hline
> 16 & 69.31 & 69.22 & 91.88 & 89.23 \\\\
> 32 &\textbf{72.56} & 96.22 & 91.85 & 89.18 \\\\
> 64 &72.20 & \textbf{96.33} & \textbf{91.91} & \textbf{89.30} \\\\
> 128 & 61.01 &96.22 & 91.87  & 89.25 \\\\
> \\hline
> \\end{array}
> >\\begin{array}{|c|c|c|c|c||}
> \\hline
> \text{BERT (num. of subspace)} & \text{RTE} & \text{SST2} & \text{QQP} (Acc.) & \text{QQP} (F1) \\\\
> \\hline
> 16 & 60.65 & 92.09 & \textbf{90.53} & \textbf{87.32} \\\\
> 32 & 62.09 & 92.09 & 90.46 & 87.22 \\\\
> 64 & \textbf{63.18} & 91.74 & 90.42 & 87.18 \\\\
> 128 & 62.82 & \textbf{92.2} & 90.49 & 87.27 \\\\
> \\hline
> \text{RoBERTa (num. of subspace)} & \text{RTE} & \text{SST2} & \text{QQP} (Acc.) & \text{QQP} (F1)\\\\
> \\hline
> 16 & \textbf{71.12} & \textbf{96.33} & 91.85& 89.22  \\\\
> 32 & 70.76 & 95.87 & 91.77 & 89.10 \\\\
> 64 & 67.51 & 96.10 & \textbf{91.87} & \textbf{89.23} \\\\
> 128 & 68.23 &95.87 & 91.81 & 89.11 \\\\
> \\hline
> \\end{array}
> The results show that the model is more **consistently sensitive to the subspace dimension** than to the number of subspaces. In particular, a subspace dimension of 64 achieves the best performance across most cases.
> In contrast, the optimal number of subspaces is less consistent across tasks and model backbones, suggesting that this choice depends on both the architecture and the dataset. **We add these results to Table 13 in the revised paper (page 21)**.
>
> - Ans. of Q3.
> > We add 'azimuth' to Figure 2 in the revised paper (page 5).
>
> - Ans. of L.
> > We address this point in our response to W1.

---

> > ### Author Rebuttal · Reviewer_93MA · 2026-04-02
> >
> > My questions have been resolved, and I am inclined to increase my score to 5 (accept).

---

### Decision · Program_Chairs · 2026-04-30

**Decision:**

Accept (regular)

**Comment:**

Reviewers agreed that the paper is well written, presents a useful and novel method, and positively highlighted the extensive and diverse experimental evaluation. Initial concerns were addressed adequately during the rebuttal phase. Hence, the paper will be a good addition to the conference program.